# FAIRNESS FEEDBACK LOOPS:
# TRAINING ON SYNTHETIC DATA AMPLIFIES BIAS

## ABSTRACT

Model-induced distribution shifts (MIDS) occur as previous model outputs pollute new model training sets over generations of models. This is known as *model collapse* in the case of generative models, and *performative prediction* or *unfairness feedback loops* for supervised models. We provide a taxonomy for MIDS and demonstrate that their fairness effects lead to a lack of representation and performance on minoritized groups within a few generations. We improve upon this unfairness behavior by situating Algorithmic Reparation as an intentional MIDS with the goal of providing redress for historical discrimination and improving the fairness of models subject to other MIDS. Our work makes an important step towards identifying and mitigating the justification of unfair feedback loops via the algorithmic objectivity and idealism ascribed to autonomous systems.

## 1 INTRODUCTION

Negative fairness feedback loops have posed problems for both machine learning practitioners' models and societies' policies for some time. One example are the 1930s Home Owner Loan Corporation (HOLC) Security Maps rediscovered by historian Kenneth T. Jackson in the 1980s. These depict 'redlining,' where minoritized communities (especially Black and Jewish people) were discriminated against in housing in America (Administration, 1938; Nelson et al., 2020; Jackson, 1985). These maps were likely used by government and banks to determine which neighborhoods should be provided programs and loans, feeding a feedback loop of segregation, limited Black home ownership, environmental racism, and increasing median household income gap between Black and white families (Shkembi et al., 2022; Einhorn & Lewis, 2021). More recently, automated systems are used for policies such as loan eligibility and approval prediction (Wu, 2022), risking the entrenchment of inequitable feedback loops behind a guise of algorithmic objectivity and idealism.

In the machine learning fairness community, this effect is often described as performative prediction (Perdomo et al., 2020) or fairness feedback loops (Lum & Isaac, 2016), where the error and behavior of a model influence its future inputs, causing runaway unfairness (Ensign et al., 2017). Increased attention to these effects, among others, and the proliferation of generated content on the internet, has created terminology for a dataset 'ecosystem.' These ecosystems may suffer from 'synthetic data spills,' a recent example being unrealistic AI-generated images of baby peacocks polluting and dominating their image search results (Shah & Bender, 2023). Over time, the presence of generated data polluting training sets is known as *model collapse*, where new generative models are trained on previously-generated data and so diverge from the original distribution, eventually converging to a point estimate as the lineage of models grows (Shumailov et al., 2023). However, researchers and practitioners lack a unified view of models' impacts on their data ecosystem and lack an understanding of the harms or benefits that may arise.

In this paper we introduce *model-induced distribution shift* (MIDS) to describe model-induced changes to the data ecosystem. MIDS are a subset of distribution shifts which are caused by past generations of model (mis)behaviours (in)advertently impacting successive generations. MIDS may be seen as the gradual pollution of a model's training set ecosystem from the synthetic data of its predecessors, particularly in locations such as the web where synthetic data is regularly published and re-scraped to form new training sets. We analyze the behavior and fairness impact of MIDS occurring over generations of models; including supervised models trained with labels sourced from their predecessors' predictions, and generative models trained from synthetic data created by their

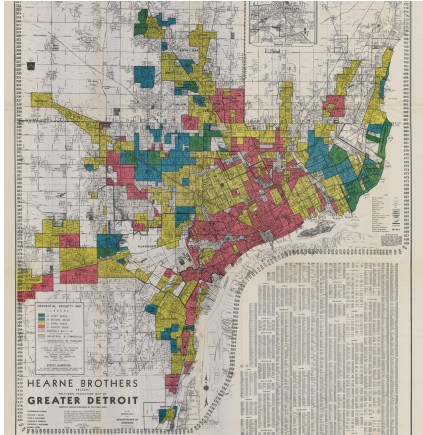 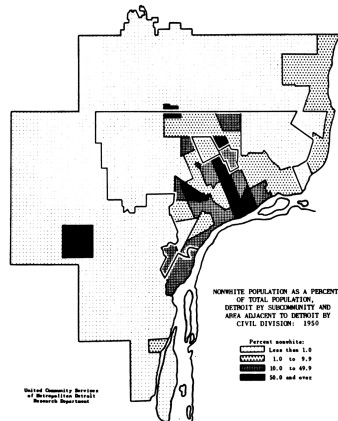

Figure 1: *Left:* 1939 Home Owner Loan Corporation Security Map for Detroit. Red areas are Grade D, or 'hazardous' locations due to the presence of racial and religious minorities (Nelson et al., 2020). Banks of the time were privy to these otherwise secret maps and in surveys stated that only high-graded neighborhoods would be offered loans (Jackson, 1985; Nelson et al., 2020). Redlining continued *de jure* until the Fair Housing Act in 1968 (90th United States Congress, 1968). *Right:* Census data on the non-white population in metropolitan Detroit from 1955 (Detroit Deomgraphics).

predecessors' outputs. We evaluate the accuracy, population demographics, demographic parity difference, and equalized odds difference of models in each generation; finding disproportionately negative impact on minoritized groups in terms of performance and representation. We find that chains of generative models eventually converge to the majority and amplify model mistakes that come to dominate and degrade the data until little information from the original distribution remains.

Our results motivate a need for work in areas such as watermarking, model attribution, model monitoring, data curation, and data provenance to mitigate the consequences of MIDS or to effectively take advantage of their benefits. For example, Figure 1 shows the HOLC Residential Security Maps for Detroit alongside demographic information of the Black population. This map itself is a historical multiclass classification model reflecting the values and priorities of the individuals and institutions responsible for its creation; specifically the white male gaze of Depression-era professional realtors (So et al., 2022; D'Ignazio & Klein, 2020). This historical context is being repurposed in more recent work in housing loan prediction to reduce discrimination through an interventionist framework called algorithmic reparation (So et al., 2022).

*Algorithmic reparation* (AR) is an amalgam of Intersectional, critical race, feminist, and queer theory to intervene and correct for oppression in automated systems (Davis et al., 2021). In this work, we reformulate AR into an intentional MIDS with the goal of reducing societal inequity and correcting for historical oppression; we use AR to reduce the unfairness impacts of other MIDS by sampling for minoritized group representation, leading to better downstream fairness over time.

In summary, we make the following contributions:

- We define a new term, model-induced distribution shift (MIDS), to unify several distribution shifts under one concept, and explore two settings to illustrate their impact. This unification draws attention to the common causes of MIDS and enables analysis even where MIDS co-occur.

- We find that MIDS can quickly lead to poor performance, class imbalance, lack of minoritized group representation, and unfairness over generations.

- We reformulate algorithmic reparation as an intentional MIDS with the goal of using model impact to effect change in the underlying distribution. We provide several studies demonstrating the ability of AR to lessen disparate impact between sensitive groups and combat the unfair effects of other MIDS.

## 2 WHAT ARE MIDS?

Several terms in existing literature describe a distribution shift caused by an amassing inheritance of polluted data over generations of models. We provide a taxonomy of these MIDS, their enablers, and their relationships to fairness. We then describe algorithmic reparation and its connection to MIDS. A background on bias, fairness in ML (FML, acronym from Davis et al. (2021)), and critiques of FML may be found in Appendix A. A full review of prior work and the MIDS and enablers discussed below is in Appendix B.

### 2.1 TAXONOMY OF MIDS

| **Taxonomy** | Pollution Location | | Enablers |
|---|---|---|---|
| Concept Drift | Outputs | Performance | |
| Label drift | Performative Prediction, Fairness Feedback Loops | | Data Annotation, Sampling |
| Input drift | Model Collapse | Disparity Amplification | Generative Feedback, Sampling |

Table 1: Taxonomy for MIDS existing in the literature as organized by the concept drift influence, either on the labels or inputs (input drift may cause label drift as well), and by the pollution location. "Outputs" refers to model outputs contaminating the training ecosystem, "performance" pollution is where the model performance influences user participation in the training ecosystem. Enablers are not, on their own, MIDS, but may complicate their effects and permit MIDS to co-occur.

Our taxonomy organizes phenomena from the literature into MIDS along two axes: the location where model impact pollutes the future distribution, and whether the effect yields concept drift in labels or data inputs. In pollution from model outputs, generated samples or annotations leak into datasets which may cause new models to be biased by their predecessors. This is known as *model collapse* (Shumailov et al., 2023) for generative models, *performative prediction* (Perdomo et al., 2020) for supervised models, and *fairness feedback loops* (Green, 2020) in supervised models for the FML community. Performance pollution may cause *disparity amplification* (Hashimoto et al., 2018) where sensitive populations of users choose to disengage from the model and future data ecosystem due to inadequate model utility. The distribution shift caused by the MIDS discussed here occur as the models are updated over time and retrained on the data ecosystem their predecessors' influenced. Throughout the remainder of the paper, we refer to generations, lineages, or sequences of generative and classifier models to indicate the teacher–student model chains underlying these MIDS.

Enablers refer to other effects that may provide signal to data ecosystems undergoing MIDS. If the enabler misrepresents the distribution to a training model, this may bias its behavior and outputs. Enablers are *not* innately MIDS, but when combined with the settings of MIDS may provide signal that can exacerbate or moderate their effects. For example, a person or model providing feedback when training each generation of a lineage of generative models may influence the rate of model collapse or disparity amplification. These enablers can also permit MIDS to co-occur: a pseudo-labeling model (part of data annotation) may annotate synthetic data from generative models for supervised training, allowing model collapse and fairness feedback loops to co-occur. Note that if the enabler is learned from a data ecosystem, they too may be subject to MIDS, though this discussion is beyond the scope of this work.

### 2.2 ALGORITHMIC REPARATIONS

*Algorithmic reparation* (AR), introduced by Davis et al. (2021) provides a transdisciplinary framework to address inequity by combining critical theory and legal reparations to identify, mitigate, and rectify societal bias. Instead of aiming for (sometimes flawed) FML goals for model performance, AR leverages bias to make entire systems more just and equitable. For example, if attempting reparative predictive policing for Black communities, interventions could include lowering police presence or arrest rates in Black neighborhoods, weighting Black records to compensate for over-representation in policing, increasing the model threshold for a negative prediction, and lowering bail

or parole requirements (Humerick, 2019). These actions are not restricted to algorithmic changes; a truly reparative approach would require transdisciplinary collaboration and a shift of economic, legal, and societal incentives.

We re-model AR into a MIDS that uses model behavior to influence the underlying distribution, leading to fairness and equity in the data setting. Intuitively, this leads to fairness in the models trained from this data. As the reparative models are updated over time, they will mirror the impact of their predecessors and affect similar change downstream. In this perspective, AR is a version of system maintenance with focus on reparative justice; permitting model owners to re-evaluate reparative goals as time progresses.

## 3 METHODOLOGY

In this section we introduce two settings, sequences of classifiers and sequences of generators, to allow for the observation and evaluation of MIDS. We also position AR as an intentional MIDS aiming for justice for historical discrimination and oppression. These settings provide an understanding of model impact over many generations by enabling informed maintenance of model and data ecosystem 'health' and accountability to fairness impacts.

In both settings, we require two classifiers to act as oracles to annotate generated samples used in training and to quantify the degree of demographic shift in generated outputs. These oracles, a label and sensitive attribute classifier, $A_L$ and $A_S$, are trained from dataset $D = X \times L \times S$, where $X$, $L$, and $S$ are the data, labels, and sensitive group attributes. This approach has been used to annotate samples for the fair training of generative models, provided the oracles generalize well to the training set distribution (Li et al., 2022; Grover et al., 2019b;a). Note that $A_L$ and $A_S$ could also be human annotators conducting manual data annotation.

We also train a generative model, $G_0$, to act as an oracle for the original training distribution. This initial generator fits well for settings where synthetic data is preferred over human-generated data for a downstream task, which may arise in FML and privacy (Zemel et al., 2013; Ganev et al., 2022; NIST, 2018). Sampling from $G_0$ as opposed to the training set also allows a chance at sampling from groups that might not otherwise be well-represented in the training set, as in Zemel et al. (2013). The three oracles, $G_0$, $A_L$, and $A_S$, provide an infinite data source used to train all other models in our settings. Therefore, if the oracles misrepresent the distribution, all downstream models trained from their outputs will experience MIDS relative to the original training distribution. Note that training on oracle-produced data breaks the *i.i.d. assumption*, and will be subject to the oracle's behavior via the sampling and data annotation enablers.

### 3.1 SEQUENTIAL CLASSIFIERS

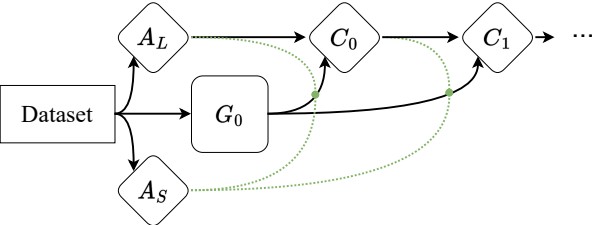

Figure 2: A high-level depiction of sequentially training classifiers (SeqClass setting) for MIDS such as performative prediction and runaway feedback loops. The green dashed lines indicate an avenue for algorithmic reparation when sampling from the generator by balancing the classifier's training set using annotations from the prior classifier and $A_S$.

The sequential classifier (SeqClass) setting permits us to pursue the study of MIDS such as fairness feedback loops and performative prediction, where concept drift is mediated by mislabeling compounding over generations of classifiers as shown in Figure 2. In the first generation, we train a classifier $C_0$ by sampling from the generator $G_0$ and labeling with oracle $A_L$. In subsequent genera-

tions $i = 1 \ldots n$, the classifier $C_i$ is trained on data sampled from $G_0$ but labeled by the proceeding classifier $C_{i-1}$. This formulation is shown below, where $\mathcal{T}_\mathcal{C}(\cdot, \cdot)$ is the classifier training algorithm sampling from its first argument and labelling from its second argument, and $\mathcal{T}_\mathcal{G}$ is the generator training algorithm. The term causing the MIDS is in red bold face.

$$C_i = \mathcal{T}_\mathcal{C}(G_i, \boldsymbol{C_{i-1}}) \text{ where } C_0 = \mathcal{T}_\mathcal{C}(G_0, A_L) \text{ and } G_0 = \mathcal{T}_\mathcal{G}(X) \tag{1}$$

## 3.2 SEQUENTIAL GENERATORS AND CLASSIFIERS

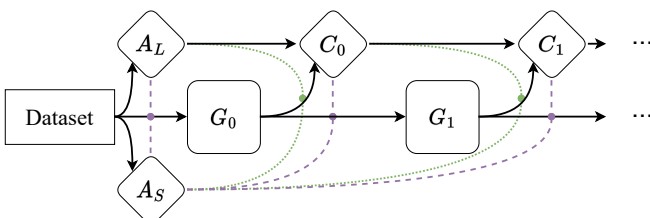

Figure 3: A high-level depiction of sequentially training generators and classifiers (SeqGenSeqClass). The narrow-dashed green lines are for classifier-side AR, and broad-dashed purple for generator-side AR. These AR methods may co-occur.

The sequential generator (SeqGen) setting allows investigation of model collapse MIDS, where distribution shift is mediated through artifacts of generated data, as shown in Figure 3. If the distribution shift from the generator training process $\mathcal{T}_\mathcal{G}$ changes the demographic distribution over time, this setting also permits an examination of disparity amplification. For this setting, we train generators $G_i$ sequentially from the samples of the proceeding generator $G_{i-1}$, where the first generator $G_0$ is trained from the original dataset. This is the same setting as used by Shumailov et al. (2023). Departing from Shumailov et al. (2023), we also train a downstream classifier $C_i$ by sampling the corresponding generator $G_i$ with labels provided from either the proceeding classifier $C_{i-1}$ or from $A_L$. The former case is the sequential generator and sequential classifier setting (henceforth SeqGenSeqClass), and the latter the sequential generator non-sequential classifier setting (SeqGenNonSeqClass). In SeqGenSeqClass, the classifiers are chained together and suffer the MIDS described in the SeqClass setting. These downstream classifiers allow us to initiate the study of downstream classifier performance and FML fairness metrics while also tracking the devolution of minoritized group representation and model collapse. In either case, the formulation for SeqGen with classifiers is shown below, with a substitute model $C$ that may stand for $A_L$ or $C_{i-1}$ depending whether classifiers are sequential. The MIDS term(s) are bolded in red.

$$G_i = \mathcal{T}_\mathcal{G}(\boldsymbol{G_{i-1}}) \text{ where } G_0 = \mathcal{T}_\mathcal{G}(X) \tag{2}$$

$$C_i = \mathcal{T}_\mathcal{C}(G_i, \boldsymbol{C}) \text{ where } C_0 = \mathcal{T}_\mathcal{C}(G_0, A_L) \tag{3}$$

## 3.3 ADDING ALGORITHMIC REPARATION

We propose using algorithmic reparation, itself a MIDS, to direct and moderate the negative influences of other MIDS. In our SeqClass setting, AR may occur at dataset collection, while training, or inferencing the classifiers. In this work we use sampling, a MIDS enabler, to guide AR to counter effects of MIDS and provide justice. Our AR approach's sampling is informed by pseudo-labelling from $C_{i-1}$ and $A_S$ to create a more equitably-balanced training batch. This is not the only avenue for AR, but is inspired by work done by the FML community for performative prediction (Ensign et al., 2017).

We add balanced sampling from $G_0$ with respect to sensitive attribute(s) and class label as part of training algorithm $\mathcal{T}_{\mathcal{A},\mathcal{C}}$. The term causing the MIDS is bolded and in red.

$$C_i = \mathcal{T}_{\mathcal{A},\mathcal{C}}(G_i, \boldsymbol{C_{i-1}}, A_S) \text{ where } C_0 = \mathcal{T}_{\mathcal{A},\mathcal{C}}(G_0, A_L, A_S) \text{ and } G_0 = \mathcal{T}_\mathcal{G}(X). \tag{4}$$

Algorithmic reparation for SeqGenSeqClass may occur at all the same points described above, with the addition of interventions taken while training the generators. In this work we examine

classifier-side AR (taken while training classifiers) and generator-side AR (while training generators). In generator-side AR, the representative and balanced batches may yield generators with balanced representations. The classifier- and generator-side AR are in Equation (5) and Equation (6), respectively:

$$C_i = \mathcal{T}_{\mathcal{A},\mathcal{C}}(G_i, \boldsymbol{C}, A_S) \text{ where } C_0 = \mathcal{T}_{\mathcal{A},\mathcal{C}}(G_0, A_L, A_S) \tag{5}$$

$$G_i = \mathcal{T}_{\mathcal{A},\mathcal{G}}(\boldsymbol{G_{i-1}}, \boldsymbol{C}, A_S) \text{ where } G_0 = \mathcal{T}_{\mathcal{A},\mathcal{G}}(X, L, S). \tag{6}$$

## 4 EVALUATION

We conduct two main sets of experiments; the first to illustrate the SeqClass setting and its relationship to fairness feedback loops, the second to show the SeqGenSeqClass setting. We find significant degradation of classifier utility and fairness due to MIDS in both settings, with worse performance in SeqGenSeqClass. We also see an accuracy gap between groups (see Figures 14 and 15), which may cause disparity amplification. In SeqGenSeqClass, we show that the beneficial class and majoritized group eventually dominate the generated samples. We lessen these unfair behaviors with classifier-side AR for SeqClass and generator-side AR for SeqGenSeqClass to create balanced representations that lead to downstream classifier fairness. We also compare SeqGenSeqClass and SeqGenNonSeqClass and find that fairness feedback loops can chase the distribution shift from model collapse and lessen its rate and performance impact. Our settings allow us to determine the degree and dynamics of harms arising from MIDS, but also show the potential for careful maintenance of the data ecosystem to lead to equitable outcomes. Note that the SeqGen setting has already been evaluated by Shumailov et al. (2023) and the fairness evaluation of generative models is a non-trivial problem (Alaa et al., 2022).

**Experimental Setup.** We provide computer vision experiments on binary class and group versions of `CelebA`, `MNIST`, and `SVHN`. Details on the datasets, model architectures, hyperparameters, and compute specifics are in Appendix C[1]. We adapt `MNIST` and `SVHN` into `ColoredMNIST` and `ColoredSVHN` by adding color for sensitive grouping and varying the distribution between group and class balance (Appendix E.1). The terms beneficial/detrimental describe the desired/undesired classes, where the majoritized group is biased to the beneficial class. For `ColoredMNIST` and `ColoredSVHN`, we choose the beneficial class as the class converged to by model collapse. For `CelebA` we provide between 5-10 generations[2], otherwise we train for 40 generations. When training each generation, all data is sampled from the prior generator and/or classifier/annotator. Oracle performances are in Table 3, generator losses in Figure 7, and discussion of classifier performances relative to its predecessor is in Appendix G.

**Classifier Fairness Metrics.** The use of classifiers in our settings permits evaluation according to traditional FML metrics. In this work, we observe the demographic parity and equalized odds differences. The former (given in Definition 1) compares the selection rates, or positive prediction rates, between groups. The latter (given in Definition 2) is the maximum between two values: the difference between the groups' true positive rates, or between their false positive rates. For these metrics, a lower value (less difference between groups) indicates more fairness. In addition to FML metrics, we also measure classifier accuracy to quantify utility.

**Algorithmic Reparation.** We also demonstrate a toy version of AR by sampling the generator evenly[3] from the four categories created by intersecting the (binary) label from $A_L$ or $C_{i-1}$ and (binary) sensitive attribute from $A_S$. These samples are used to create batches for generator or classifier training, as described in Appendix C.4. The reparation may be limited by two causes: 1) we cap the number of samples that may be drawn to form the batch yet attempt to create equal categories from an unequal dataset; and 2) the effects of MIDS. We cap the reparation sample budget to simulate the costs of conducting AR. A minoritized group that AR increases the representation of may be re-selected more often than peers in the majoritized group(s), increasing their exposure toe

---

[1]Our code is hosted anonymously here: `https://anonymous.4open.science/r/FairFeedbackLoops-1053/README.md` and will be released to GitHub if accepted for publication.

[2]The time to repeatedly train `CelebA` from scratch took around a week, and due to the cost and $CO_2$ footprint, we elected to terminate these experiments upon realization of the MIDS.

[3]In practice, the amount drawn from each category would be adapted to the goals of reparation.

mislabeling. Additionally, the higher number of generations this data is subjected to may increase the model collapse at this area of the distribution.

## 4.1 SEQUENTIAL CLASSIFIER SETTING

Our first experiment suite uses SeqClass as described in Section 3.1; MIDS occur as a classifier's predictions are used to label the next generation's classifier. We measure the effects of MIDS with classifier accuracy, demographic parity (difference in positive prediction rates), and equalized odds (considers true and false positive rates), see Definition 1 and Definition 2). These metrics measure downstream utility and fairness for classifiers trained on generated data, goals from Li et al. (2022). These are evaluated on a held-out validation set that matches the training distribution. Due to the tension between FML metrics and societal equity and fairness, we do not necessarily desire to achieve good performance so much as observe the dynamics of our settings and reparation methods. Refer to Figure 4, Figure 16, and Figure 17 for results on `ColoredMNIST`, `CelebA`, and `ColoredSVHN`.

**AR reduces performance degradation from MIDS.** In `ColoredMNIST` and `ColoredSVHN`, we observe an accuracy drop of 10% over 40 generations, a far smaller impact than observed in the SeqGenSeqClass setting discussed next. The classifier unfairness generally increased (in the case of `ColoredSVHN`, by 20-40%). AR lead to a significant gain in fairness for `ColoredMNIST`, and a marginal gain in `ColoredSVHN` (though with high variance and a 10% cost to accuracy). This difference across datasets is likely due to the resampling observed when balancing demographic categories for the AR batches. In `ColoredMNIST`, the resampled proportion decreases over time, and we observe that the categories even out. In `ColoredSVHN`, the categories are dominated by the majoritized–beneficial and minoritized–detrimental categories to a higher extent than the original distribution, and AR is unable to gain balance, likely due to misrepresentation by the oracles.

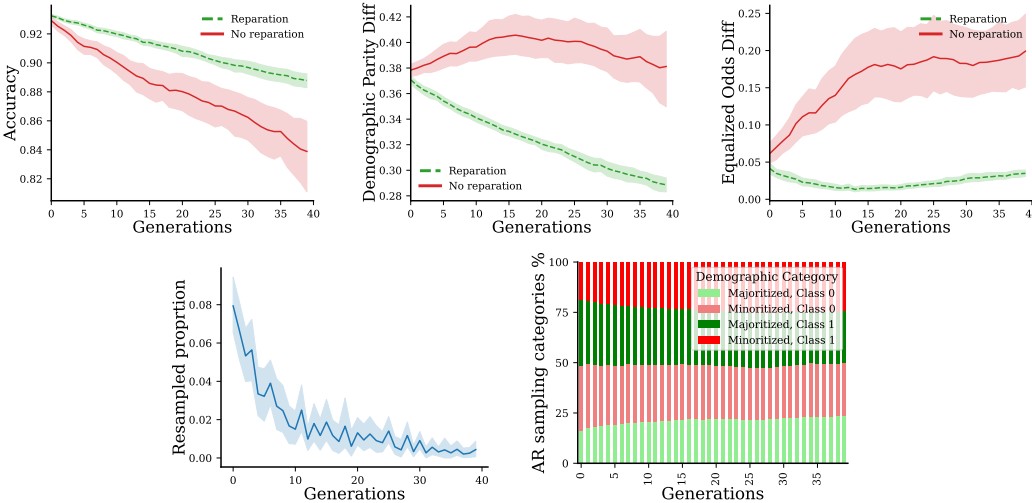

Figure 4: `ColoredMNIST` results for SeqClass on the validation set. *Top:* accuracy, demographic parity, equalized odds difference. Lower fairness indicates fairness. Better fairness is achieved with classifier-side AR. *Bottom left:* the amount of resampling from $G_0$ due to imbalanced categories after the initial sampling step indicates the batches become equally representative. *Bottom right:* the category breakdown of sampling $b + r$ times from $G_0$, as described in the initial batch creation step in Appendix C.4 and Algorithm 1.

## 4.2 SEQUENTIAL GENERATOR AND CLASSIFIER SETTING

These experiments refer to the SeqGenSeqClass setting described in Section 3.2 and depicted in Figure 3. In addition to classifier performance metrics, the changing demographics of the generators are evaluated by taking 1000 random samples and annotating their class and sensitive group balance with $A_L$ and $A_S$. Refer to Figure 18, Figure 5, and Figure 6, for the results for `ColoredMNIST`, `ColoredSVHN`, and `CelebA`.

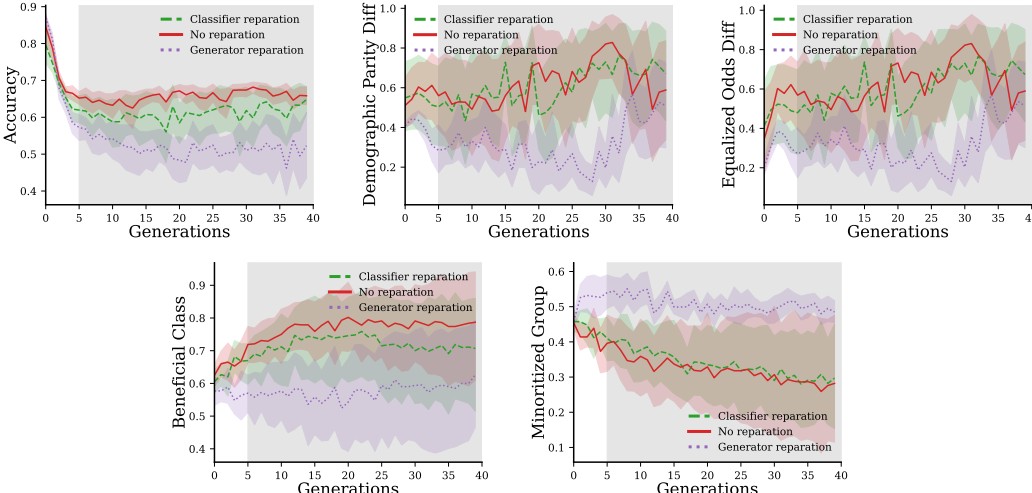

Figure 5: `ColoredSVHN` results for **SeqGenSeqClass** on the validation set, where shading shows generations with little utility due to model collapse. *Top:* shows accuracy and FML metrics (lower is better). *Bottom:* beneficial class and minoritized group representation. Generator-side AR (in purple) leads to better class and group representation and consequently more fairness, but with an accuracy cost.

If we judge model collapse as the point when the generated data ceases to have any downstream utility, model collapse occurs at generation 15 for `ColoredMNIST`, 5 for `ColoredSVHN`, and between generations 1-5 for `CelebA`. These values correspond to the increasing difficulty of the datasets' tasks, which is correlated with heavy-tailedness in their distributions (Meng & Yao, 2023). A small sample size may be able to represent a concentrated distribution, but finite samples of a heavy-tailed distribution will likely be biased, leading to faster distribution shift. This steep rate of decline into stability at 50% or random accuracy is likely due in part to the amount of synthetic data used to train each generation. As the amount of synthetic training data decreases, so to does the rate of accuracy decline and beneficial class dominance, as shown in Appendix E.2.

`ColoredMNIST` and `ColoredSVHN` contain similar images, but either due to differences in digit representation or the difference in simplicity between `MNIST` and `SVHN`, they collapse very differently (see Figure 19). `ColoredMNIST`, which has perfect class balance, eventually creates samples resembling an '8,' or all the digits superimposed. While `ColoredSVHN` collapses much faster, it generates samples that look like a '3.' The datasets also experience different trends in loss between $G_i$ and $G_{i-1}$ (Appendix D). Intuition would suggest that this value decreases constantly as model collapse simplifies the distribution and task, but for `ColoredMNIST` and `CelebA` the losses increase. This may be due to hyperparameter instability in generations, or perhaps finite sampling of the distribution lead to enough noise and bias to complicate the representation task.

**Model collapse leads to minoritized group erasure.** Model collapse, as was hypothesised by Shumailov et al. (2023), leads to effects similar to disparity amplification, but instead of minoritized groups leaving the training ecosystem due to bad performance, they are distorted and erased from the generator's representations. While early generations of `ColoredMNIST` and `ColoredSVHN` have group balance, `CelebA` has a 28% minoritized group population which is far lower than its original distribution balance of 41.6%, a similar trend is seen for `CelebA`'s class balance as well. Additionally, the performances and fairness of $C_0$ are immediately worse than $A_L$, demonstrating the immediate impact of MIDS propagating from the errors of the oracle models. As MIDS progress, the group balance of `ColoredMNIST` remains close to its initial balance, though `ColoredSVHN` steadily decreases, and `CelebA` even falls to 0% minoritized group representation.

**Performative prediction combined with model collapse yields higher accuracy.** We also uncover co-operation between MIDS by evaluating the role of sequential classifiers in **SeqGenSeqClass** and **SeqGenNonSeqClass** (see Appendix E.3). Concept drift among sequential classifiers allows $C_{i-1}$ to provide meaningful labels for training $C_i$ from $G_i$. For the non-sequential classifiers,

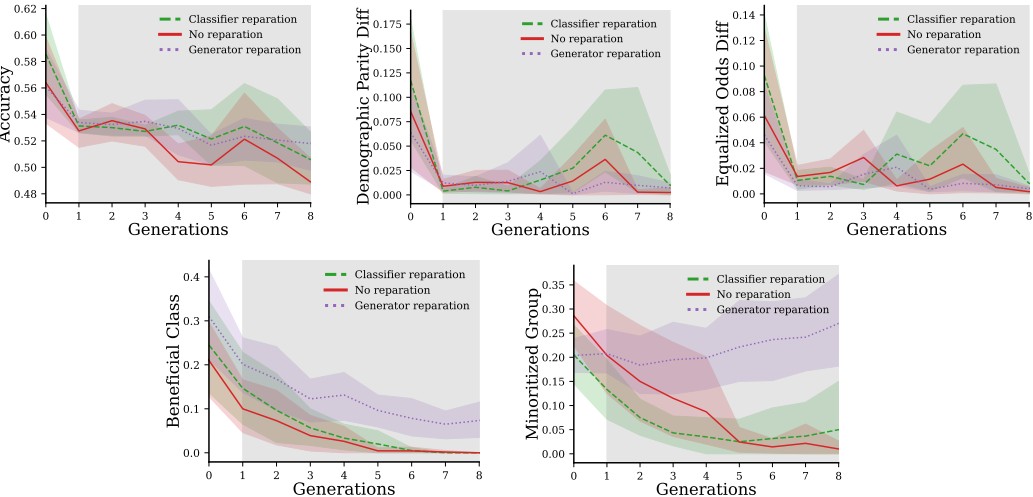

Figure 6: `CelebA` results for **SeqGenSeqClass** on the validation set, where shading shows collapsed generations. *Top:* shows accuracy and FML metrics (lower is better). *Bottom:* beneficial class and minoritized group representation. Note that the beneficial class is a minority population in the original dataset, and decreases here, though generator-side AR maintains sightly more class and far more group balance than classifier-side AR and the results with no reparation.

$A_L$ cannot adequately support the distribution represented by $G_i$ once the $i^{th}$ distribution substantially differs from the original. The inherited knowledge of $\mathbb{P}(Y|X)$ passed through the sequential classifiers allows them to preserve a more accurate map from the changing distribution to the classes.

**Generator-side AR improves fairness and minoritized representation.** Between the two AR methods we evaluated, the generator-side AR lead to more preservation of the group and label balance in all three datasets (especially in `CelebA`). This result fits intuitively as the biased sampling would enable these generators to keep a balanced representation without succumbing to disparity amplification. Generator-side AR also enabled better fairness scores in `ColoredMNIST` and `ColoredSVHN`, though at a cost to accuracy. The classifier-side reparation did not show consistent performance across datasets, achieving worse or equivalent performance to the non-reparative results, likely due to the strength of the model collapse MIDS.

## 5  CONCLUDING REMARKS

In this paper, we introduced and taxonomized model-induced distribution shifts (MIDS) and created empirical settings enabling the evaluation and moderation of their harms. With these settings we found that MIDS, both on their own and co-occurring with enablers such as data annotation, lead to major degradation in utility, fairness, and minoritized group representation. Focused on addressing the degradation to fairness, we reformulated algorithmic reparations (from the literature of critical theory in ML) into an intentional MIDS supporting holistic notions of equity and justice. This allowed us to lessen particular harmful effects from MIDS, yet at a cost to downstream accuracy.

The general impact of our findings is an indication that MIDS are of imminent concern for data ecosystems undergoing synthetic data spills, demonstrating a need for prevention. One solution is an archival perspective on data curation as introduced in Jo & Gebru (2020). Specifically, adopting the tenets of archival description codes could enable gathering of high-quality provenance information and stress the moral obligation motivating many archives' raison d'être to identify and repair structural and historical bias (DACS; RAD; Jo & Gebru, 2020). Another solution, motivated by our results, is the importance of non-synthetic data and human data annotation to prevent or slow the rate of MIDS. We therefore advocate for more attention to the often-unseen and underappreciated labor of human data workers. We end on a call for safer conditions for data workers given their current and increasing importance in our data ecosystems.

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

# A    FAIRNESS IN MACHINE LEARNING (FML)

There are several sources of bias and oppression that may be encoded into a model as a result of the processes for gathering and encoding data, training, evaluating, and deploying the model. There are also several harms these systems may commit, including allocative (where resources are withheld from certain groups, such as in redlining) and representational (where groups are stigmatized and stereotyped). There are many different biases that may co-occur such as historical, representational, measurement, aggregation, learning, evaluation, and deployment biases (Suresh & Guttag, 2021). These often arise from misrepresenting a complex feature (e.g., treating gender or sex as a binary), mis-measuring features, stripping data of its context (e.g., regional or dialectal language heteroglossia), and from historical oppression influencing the data modelling processes. There are several frameworks for defining and addressing issues of fairness; calibration (used when the sensitive identities have impact on the decision task), anti-classification (used when sensitive data is unavailable or illegal to use), individual fairness ("similar individuals should be treated similarly"), and classification parity.

In group fairness, protected attributes are often chosen from legally-protected attributes such as race or gender, and encoded into categorical features to determine *sensitive groups*. In this paper we use the terms *majoritized* and *minoritized* as in D'Ignazio & Klein (2020) to emphasize the impact of a model's behavior on a group. Note that the majority population might not correspond with the majoritized (benefit-receiving) group; for example the Black population is a minoritized majority in the COMPAS dataset(Department, 2016). This grouping often splits the dataset into two groups, delineated by one attribute with two possible values (e.g., 'male' vs 'female' or 'white' vs 'people of color') which may misrepresent or inappropriately group populations and ignores the compounding impact of possessing multiple marginalized identities.

In classification parity, there are a variety of metrics that aim for some equality of rates between these groups, such as accuracy, positive selection rate, or error rates. In this work, we use accuracy difference, demographic parity difference, and equalized odds difference to cover a multitude of differing priorities model owners may value. It is often impossible to satisfy multiple fairness metrics simultaneously, so they are ideally chosen based upon the task. The binary classification and binary grouping versions of these metrics are presented below:

**Definition 1 (Demographic Parity (DP) (Calders et al., 2009))** *A classifier $\hat{Y}$ satisfies Demographic Parity with respect to the sensitive attribute $s$ if:*

$$\mathbb{P}(\hat{Y} = 1|s = 0) = \mathbb{P}(\hat{Y} = 1|s = 1) \qquad \forall 0, 1 \in s.$$

In this work we consider demographic parity difference, which is the absolute value of the difference between the two terms equated above. Each term is also the selection rate, or rate of positive prediction, for the group.

**Definition 2 (Equalized Odds (EOdds) (Hardt et al., 2016))** *A classifier $\hat{Y}$ satisfies Equalized Odds with respect to the sensitive attribute $s$ if for ground truth $L$:*

$$\mathbb{P}(\hat{Y} = 1|L = l, s = 0) = \mathbb{P}(\hat{Y} = 1|L = l, s = 1) \qquad \forall l \in \{0, 1\}, \forall 0, 1 \in s.$$

We also used equalized odds difference. This is formulated as $\max[|\mathbb{P}(\hat{Y} = 1|L = 0, s = 0) - \mathbb{P}(\hat{Y} = 1|L = 0, s = 1)|, |\mathbb{P}(\hat{Y} = 1|L = 1, s = 0) - \mathbb{P}(\hat{Y} = 1|L = 1, s = 1)|]$, or the larger of the absolute value differences between the false and true positive rates for the groups.

Due to the biases that may exist in data collection, training, evaluation, and deployment, adherence to or achievement of any of these fairness metrics does not guarantee fairness or equity. This is a byproduct from FML's reliance upon algorithmic idealism, where computation assumes a meritocratic society where equalizing demographic disparities must therefore lead to fairness at a societal level (Davis et al., 2021; Green, 2020). Additionally, these fairness metrics assume that the cost borne by an error made on a sample of either group is the same, when in practice the cost and benefit may differ greatly depending on identity. FML may also engage in or reinforce two main biases: 1) automation bias, that machines are objective and are less biased than humans, and 2) that automation invites justice without regard for the objective and purpose of the models (Davis et al., 2021; Green, 2020).

# B  MIDS IN LITERATURE

We provide a detailed review of the MIDS and enablers described in our taxonomy in Section 2, as well as overview prior work in Appendix B.3.

## B.1  MIDS

**Performative prediction and runaway feedback loops.** *Performative prediction* is a distribution shift induced by training upon the predictions of a previous classifier (Perdomo et al., 2020). The fairness community has studied performative prediction and feedback loops in the context of risk assessment systems, including mortgaging and predictive policing (Green, 2020). For example, an online training model supplying policing locations may converge towards over-policing low-income non-white communities (Lum & Isaac, 2016; Richardson et al., 2019). Theoretical work has found that the degree of runaway feedback may be moderated with careful training set weighting, but cannot be negated entirely (Ensign et al., 2017).

**Model Collapse.** Where performative prediction and runaway feedback loops generally refer to classification models, model collapse describes the same effect for generative models. *Model collapse* occurs when new generative models are trained on samples created by their predecessor over many generations, as introduced in Shumailov et al. (2023) and concurrently in Alemohammad et al. (2023). This leads to new models forgetting the original data distribution as they recreate and amplify the failures of their ancestors. There are two error sources that contribute: 1) functional approximation error due to an inadequately expressive generator, and 2) statistical approximation error from finite sampling. Model collapse begins with a loss of information from the tails of the data distribution. In late-stage model collapse, the model mixes the modes of the original distribution, converging to a point estimate of some mean betwixt them. There are also concerns over the effects of model collapse on fairness, as model failure on "low-probability events" may have negative effects on minoritized groups when datasets have poor representation (Suresh & Guttag, 2021).

**Disparity amplification.** Unlike the aforementioned MIDS, disparity amplification arises from human-model interaction. If a model suffers from problems derived from representational bias, it may have overall high performance but low performance on minoritized groups. This can lead to *disparity amplification*, where minoritized users who suffer high error rates may choose to disengage from the model, shifting the future dataset towards the majoritized group (Hashimoto et al., 2018).

## B.2  MIDS ENABLERS

We describe several enablers in more detail here. Note that even sampling is an enabler, and indeed it informs our approach to algorithmic reparation in our experiments.

**Pseudo-labelling.** *Pseudo-labelling* generally refers to using a model to assign labels to unlabeled samples in a dataset, so that this data may also be used for supervised or semi-supervised training (Cascante-Bonilla et al., 2020). This may occur just once (Cascante-Bonilla et al., 2020) or iteratively (McLachlan, 1975). Incentives for pseudo-labeling may arise in cases where manual labeling and/or annotation is too expensive for vast swathes of data. The fairness impact of using pseudo-labelling for self-supervised learning was discussed in Zhu et al. (2022), finding that groups with high initial accuracy benefit whereas groups with low initial accuracy may see a degradation in performance.

**Feedback and Data Annotation.** Similarly to pseudo-labelling, feedback (whether human or model-based) is often used for labeling and annotating data for supervised training or for providing feedback on generative outputs. Reliance on human annotation can lead to unfairness arising from individual annotator bias and instructions for annotating (Suresh & Guttag, 2021). Recent work has found indicators that some human data annotators use LLMs or other models, which may lead to MIDS if these models are updated and retrained on the data they labeled (Veselovsky et al., 2023). AI feedback is also used in methods such as Constitutional AI, which uses a succession of fine-tuned supervised models to provide RLAIF (reinforcement learning from AI feedback) for training 'harmless' AI assistants (Bai et al., 2022).

| ColoredMNIST | Class | | ColoredSVHN | Class | |
|---|---|---|---|---|---|
| Group | Beneficial | Detrimental | Group | Beneficial | Detrimental |
| Majoritized | 0.350 | 0.150 | Majoritized | 0.424 | 0.118 |
| Minoritized | 0.150 | 0.350 | Minoritized | 0.183 | 0.275 |

| CelebA | Class | |
|---|---|---|
| Group | Beneficial | Detrimental |
| Majoritized | 0.187 | 0.396 |
| Minoritized | 0.300 | 0.116 |

Table 2: Class and group demographics of training datasets. Values show the proportion of that group–class category in the training dataset (and therefore sum to 1). *Top:* ColoredMNIST and ColoredSVHN the majoritized is group skewed towards the beneficial class with probability 0.7, and to the detrimental class with probability 0.3. *Bottom:* CelebA.

## B.3 RELATED WORK

We overview three pertinent related works that study MIDS and how they connect to our work. Two works, Shumailov et al. (2023) and Alemohammad et al. (2023) both study model collapse (relevant to SeqGen). The former includes a toy example of predicting the mean of a one-dimensional Gaussian over while undergoing model collapse, finding that sampling must increase quadratically over time to remain accurate against the sampling and functional approximation error.

Another work, Taori & Hashimoto (2023), finds a theoretical upper-bound for the amplification of bias (difference between a metric in the current generation compared to the original distribution) due to model feedback (relevant to SeqGenNonSeqClass and SeqClass). They provide two interventions to stabilize bias amplification, by adding non-synthetic data (see Appendix E for an ablation of this) and by overfitting the models. We primarily study their worst-case scenario where there is no additional non-synthetic data to stabilize the MIDS. Additionally, our investigation of bias departs from calibrating models with respect to the original dataset distribution, we use algorithmic reparation to intervene in unfair systems achieving a fairness ideal that may be different from the original dataset. We also expand upon their work by considering the cooperation between co-occurring MIDS as in our SeqGenSeqClass setting.

## C EXPERIMENTAL DETAILS

### C.1 DATASETS

We evaluate the fairness effects of model collapse and algorithmic reparation on several datasets; CelebA, and adapted versions of MNIST and SVHN. Throughout our experiments, we treat class 1 as the 'beneficial' class.

**ColoredMNIST** MNIST is a single-channel handwritten digit recognition dataset (Deng, 2012). We use 50000 images for training, 10000 for validation, and another 10000 for testing. We adapt MNIST to a binary classification scheme (determining if a digit is in [0..4] for class 0 or [5..9] for class 1). The class label is switched with a uniform probability of 5% to add label noise, as in Arjovsky et al. (2020). We also adapt MNIST to have binary groups by coloring the sample either red or green, where green is treated as the 'majoritized' group. We skew the majoritized group to the beneficial class, such that $\mathbb{P}(S = \text{majoritized}|L = \text{beneficial}) = 0.7$, $\mathbb{P}(S = \text{majoritized}|L = \text{detrimental}) = 0.3$. In this case, both classes and groups are balanced, as seen in the dataset composition matrix in Table 2. The testing set follows a slightly different distribution where there is no skew between group and class, i.e., $\mathbb{P}(S = \text{majoritized}|L = \text{beneficial}) = \mathbb{P}(S = \text{majoritized}|L = \text{detrimental}) = 0.5$. Ablations for this skew and class and label balances are in Appendix E.

**ColoredSVHN** SVHN (Street View House Numbers) is a digit recognition dataset composed of house numbers sourced from Google Street View (Netzer et al., 2011). We use 52327 images for training, 20930 for validation, and another 26032 for testing. For SVHN, we adapt to a binary task similarly as in MNIST. We binarize the classification task to determining if a digit is in [0..4] for class 1 or [5..9] for class 0. The class label is swapped with a uniform probability of 5% to add label noise, as in Arjovsky et al. (2020). Unlike in ColoredMNIST, class 1 is the lower numbers as SVHN converges to small numbers over the course of model collapse, as seen in Figure 19. This causes class imbalance; class 1 composes 60.7% of the data. We also add sensitive groups by converting the images to grayscale and then coloring the samples either red or green as in ColoredMNIST. The green group again serves as the majoritized group, and is skewed towards the beneficial class at rates $\mathbb{P}(S = \text{majoritized}|L = \text{beneficial}) = 0.7$, and $\mathbb{P}(S = \text{majoritized}|L = \text{detrimental}) = 0.3$, leading to group imbalance with the majoritized group as 54.3%. A matrix showing the composition of the ColoredSVHN training distribution is shown in Table 2. The testing set follows a slightly different distribution where there is no skew between group and class, i.e., $\mathbb{P}(S = \text{majoritized}|L = \text{beneficial}) = \mathbb{P}(S = \text{majoritized}|L = \text{detrimental}) = 0.5$, leaving only the effect of the class imbalance.

**CelebA** CelebA is a dataset of celebrity faces (Liu et al., 2015). We use an 80/10/10 train/validation/test split of the 202599 cropped and aligned images. The binary classification task is to predict attractiveness, where sensitive groups are given from gender ('Male,' 'not Male'). The group and class balance is 58.% 'not male' and 51.3% 'attractive.' The composition of the dataset is shown in Table 2. This dataset is criticized for the inclusion of subjective features such as 'attractive,' and there are many instances of incorrect labeling and annotation (Lingenfelter et al., 2022). In the other datasets, we chose the beneficial class as the class most converged to by model collapse. Interestingly, through towards the class imbalance, model collapse converges to 'unattractive,' which would benefit the 'Male' group more. However, model collapse also converges to 'Not Male.' We therefore use 'attractive' as the beneficial class and 'Not Male' as the majoritized group.

**Justification of dataset choice.** We choose ColoredMNIST and ColoredSVHN due to their similarity. Both detect and classify digits, and may be easily adapted into a binary classification and binary fairness grouping task. They differ in their complexity, SVHN is usually the harder dataset to learn, and also in their class balance once binarized (see Table 2). These two datasets, despite their similarities, show very different points of model collapse and even opposite behavior when observing the accuracy difference between groups, as in Figure 14 and Figure 15. CelebA is chosen to represent a more complex and real-world dataset with well-documented disparities between the class and grouping we use for its task (Lingenfelter et al., 2022). All of these datasets, when used as binary grouping and binary classification problems, enable us to effectively observe our chosen utility and fairness metrics.

## C.2 COMPUTE AND CODEBASE

Experiments were performed on Ubuntu 18.04.6 LTS using 4 Intel Xeon CPU cores per GPU. We use the following GPUs per random seed: for ColoredMNIST we use 1 NVIDIA T4 GPU with 16 gigabytes of memory; for ColoredSVHN we use 2 NVIDIA RTX6000s with 40 gigabytes each; for CelebA we use 2 NVIDIA A100s with 40 gigabytes each. Our codebase is in Python 3.9 with PyTorch and fairness metrics from Fairlearn (Bird et al., 2020). Existing code was adapted for our experiments: VAE models from Subramanian (2020), ColoredMNIST from Arjovsky et al. (2020), SVHN with fairness from Kenfack et al. (2021), ResNets from Idelbayev (2018).

## C.3 MODELS

The architectures and hyperparameters differ based on dataset. The performance of the annotator models $A_L$ and $A_S$ are shown in Table 3. All generative models were trained with ADAM (weight decal $1 \times 10^{-5}$), and all classifiers with SGD and cross-entropy loss.

**ColoredMNIST**

- Classifiers and annotators are 6 layer CNNs with 2 convolution layers and ReLU activations. Learning rate 0.1, batch size 256, for 30 epochs.

- Generators (VAEs) are mirrored encoder and decoder CNNs. Each is 2 convolution layers with ReLU activations. Uses BCE Loss with KL-divergence term, a latent space dimension of 20, and a variational beta of 1. Learning rate 0.001, batch size 256, for 30 epochs.

**ColoredSVHN**

- Classifiers and annotators are 32-layer ResNets adapted from Idelbayev (2018). Learning rate 0.001, batch size 32, for 30 epochs.

- Generators are the deep convolutional VAE adapted from Sujit (2019), using MSE loss with KL-divergence term, a latent space dimension of 32, and a variational beta of 1. Learning rate 0.0005, batch size 128, for 30 epochs.

**CelebA**

- Classifiers and annotators are 110-layer ResNets adapted from Idelbayev (2018). Learning rate 0.001, batch size 128, for 15 epochs.

- Generator VAEs are composed of a 5-layer CNN encoder and 6-layer upsampling CNN decoder with LeakyReLU activations. Loss is BCE with KL-divergence term, a latent space dimension of 500, with a variational beta of $5 \times 10^{-6}$. Learning rate 0.005, batch size 64, for 30 epochs.

Table 3: Performances (with standard deviations) of the label and sensitive attribute annotator models $A_L$ and $A_S$. Performances are shown for each dataset. The performance of $A_S$ is high for ColoredMNIST and ColoredSVHN as determining sample color is an easy task. The fairness metrics for $A_S$ should be close to 1, as these models should assign class based on the sensitive attribute alone.

|  | ColoredMNIST | | ColoredSVHN | | CelebA | |
| --- | --- | --- | --- | --- | --- | --- |
|  | $A_L$ | $A_S$ | $A_L$ | $A_S$ | $A_L$ | $A_S$ |
| Accuracy | $0.928 \pm 0.003$ | $1 \pm 0$ | $0.849 \pm 0.080$ | $1 \pm 0$ | $0.816 \pm 0.005$ | $0.976 \pm 0.002$ |
| $\Delta$ Accuracy | $0.009 \pm 0.005$ | $0 \pm 0$ | $0.052 \pm 0.111$ | $0 \pm 0$ | $0.029 \pm 0.008$ | $0.015 \pm 0.008$ |
| $\Delta$ DP | $0.367 \pm 0.008$ | $1 \pm 0$ | $0.151 \pm 0.193$ | $1 \pm 0$ | $0.440 \pm 0.022$ | $0.951 \pm 0.004$ |
| $\Delta$ EOdds | $0.032 \pm 0.022$ | $1 \pm 0$ | $0.163 \pm 0.240$ | $1 \pm 0$ | $0.271 \pm 0.037$ | $0.971 \pm 0.008$ |

### C.4 ALGORITHM FOR ALGORITHMIC REPARATION

These reparation batches may be used for training the classifiers or generators (or both). We repeat this sampling procedure at each batch with Algorithm 1, where $r$ is the reparation budget (we set this at 25% of $b$, the batch size).

---

**Algorithm 1:** Training with algorithmic reparation batches.

**Input:** Sample-providing generator $G$, batch size $b$, reparation budget $r$, label annotator $C$ (either $C_{i-1}$ or $A_L$), sensitive attribute annotator $A_S$.

**Output:** Reparation batch

1: **for** batch in number batches **do**
2:     Ideal $= [b/4, b/4, b/4, b/4]$                    ▷ Ideal category sizes
3:     Batch $= [b_{L=0,S=0}, b_{L=0,S=1}, b_{L=1,S=0}, b_{L=1,S=1}] = [0, 0, 0, 0]$    ▷ Initialize batch categories
4:     Temporary batch $=$ Sample $b + r$ times from $G$        ▷ Initial batch from uniform sampling
5:     Annotate temporary batch using $C$ and $A_S$
6:     Categorize batch depending on $L$ and $S$ values from annotations
7:     Populate Batch until Ideal$_i$ $=$Batch$_i$
8:     To_resample $=$sum(Ideal $-$ Batch)        ▷ Get amount to sample to fill deficient categories
9:     Batch.append(Sample To_resample times from $G$, annotate with $C$ and $A_S$)     ▷ Refill batch
10: Update model on Batch.
11: **return** Batch

---

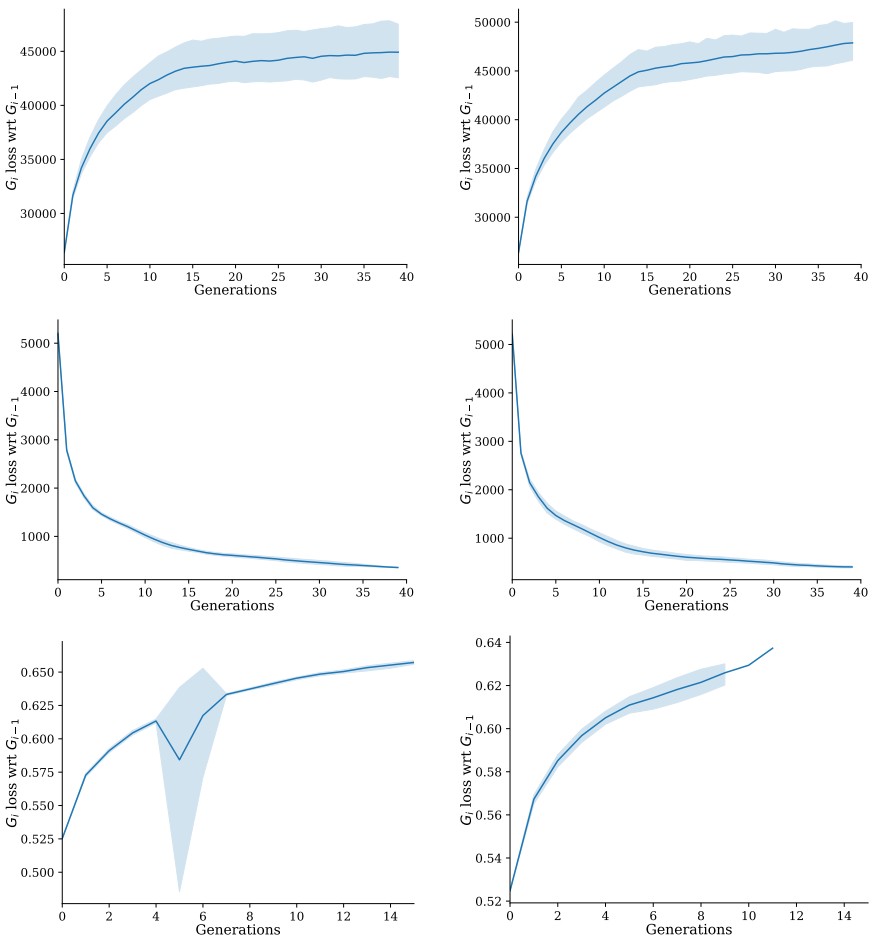

Figure 7: *Top row*: ColoredMNIST, *Middle:* ColoredSVHN, *Bottom:* CelebA. *Left*: model collapse losses, *Right:* model collapse with generator-side algorithmic reparation losses.

# D  MODEL COLLAPSE IN GENERATORS

We show the losses with respect to the parent generator loss ($\mathcal{L}(G_i, G_{i-1})$) over generations for each dataset for the normal model collapse result and with generator-side AR (which causes a minor adjustment). Intuition would suggest that the distribution collapses to be increasingly easy-to-learn, such that successive generators inherit simplified versions (due to finite sampling of their parents) of the problem and so perform better. We observe this effect with the smoothly decreasing loss curves of ColoredSVHN in Figure 7.

However, for both ColoredMNIST and CelebA, we see the exact opposite curve. The child generators are faced with an increasingly hard-to-learn distribution. We hypothesize that this may be due to one of two causes. 1) We do not perform hyper-parameter tuning for the generators at each generation, and perhaps ColoredMNIST and ColoredSVHN experience hyperparameter instability. 2) Perhaps this is simply a quirk of model collapse, finite sampling of heavy-tailed distributions may lead to enough bias and noise to significantly complicate the learning task. We propose to investigate the stability of model collapse in future work.

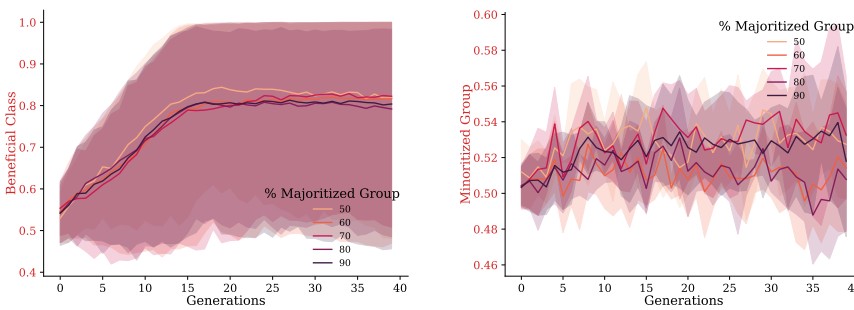

Figure 8: `ColoredMNIST` class and group balance while varying the group balance in SeqGenSeqClass.

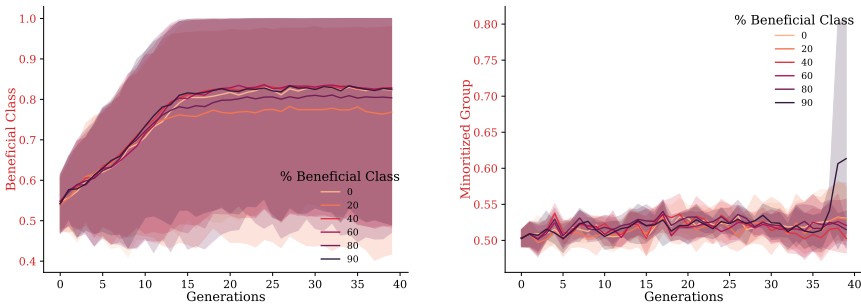

Figure 9: `ColoredMNIST` class and group balance while varying the class balance in SeqGenSeqClass.

# E APPENDIX: ABLATION STUDIES

In this appendix we provide experiments to demonstrate the effects of MIDS over several ablated variables: the sensitive group imbalance, class imbalance, and amount of synthetic training data. We provide results for both `ColoredMNIST` and `ColoredSVHN`.

## E.1 CLASS AND GROUP IMBALANCE

In these studies we varied the class balance or group balance. The study was carried out on `ColoredMNIST` with 5 seeds in the sequential generator and classifier setting. For group imbalance, the groups were equally likely to belong to the beneficial class. For class imbalance, the majoritized group was skewed towards the beneficial class in the same manner as discussed in Appendix C.1. For this task, the variations in balance did not effect the generated population or downstream classifier performance much. The generator class and group balances are shown for varied group balance in Figure 8 and for varied class balance in Figure 9. The results in Section 4.2 use datasets with a mixture of class and group imbalance which better elucidate the effects of MIDS.

## E.2 AMOUNT OF SYNTHETIC DATA

In this study we varied the amount of original training data (drawn from the training set) in each batch for training generators in the sequential generator and classifier setting. These experiments were carried out on `ColoredMNIST` for 5 seeds. There is a substantially higher accuracy cost and accuracy disparity between groups, as shown in Figure 10. Note that even with 0% synthetic data (i.e., training each generator from the original training set) there is still an accuracy loss over time due to the effects of the sequential classifiers. While the group balance is not hugely effected, the class balance skews towards the beneficial class over the generations, fueling an increase of equalized odds difference with more synthetic data, see Figure 11.

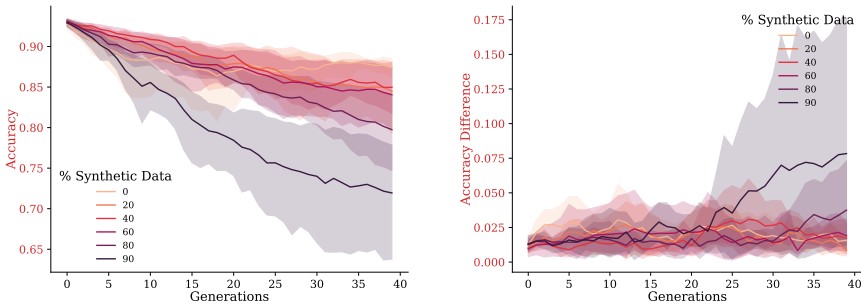

Figure 10: `ColoredMNIST` accuracy and accuracy difference (between groups) while varying the amount of synthetic data in **SeqGenSeqClass**.

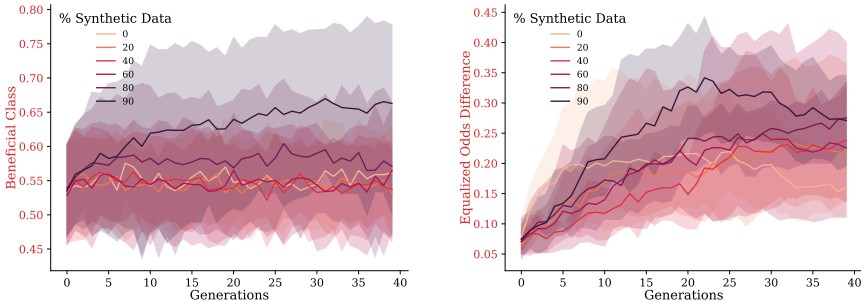

Figure 11: `ColoredMNIST` class balance and equalized odds difference while varying the amount of synthetic data in **SeqGenSeqClass**.

In practice, there may be several generations of synthetic data present when drawing from a corpus of polluted data. For example, when training $G_2$, samples from $G_0$ and $G_1$ might also be present. In this case, the compounded artefacts of model collapse will be lesser in these early generations. In this study, the synthetic data is only pooled from the most recent generator, and so these results may overstate the effect of model collapse in the aforementioned case.

### E.3 SEQUENTIAL VERSUS NON-SEQUENTIAL CLASSIFIERS IN SEQGENSEQCLASS AND SEQGENNONSEQCLASS

In this study we demonstrate the impact of sequential classifiers in **SeqGenSeqClass**. These experiments were conducted for `ColoredMNIST` and `ColoredSVHN` for 25 and 10 seeds, respectively.

Result show that the non-sequential classifiers are more sensitive to changes in the distribution, as seen in the accuracy over generations and selection rate graphs in Figure 12 and Figure 13 for `ColoredMNIST` and `ColoredSVHN`, respectively. This is consistent with the intuition that the sequential classifiers are learning as model collapse progresses, which is supported by the higher selection rate (rate of predicting the beneficial class) for the sequential models as the samples created after several generations resemble images from the original distributions beneficial class. Incidentally, for these datasets, the higher selection rates for non-sequential classifiers was not dependent on the grouping, leading to higher demographic parity and equalized odds fairness.

Note that the achievement of higher fairness metrics in the non-sequential classifier case means that there is higher fairness with respect to the original distribution. This may be undesirable in some cases, particularly those applicable to Algorithmic Reparation, which specifically notes that equality of of model outputs to base rates in a dataset will not guarantee equity if the dataset is collected with any bias, including compounding Intersectional biases which these experiments do not inform upon Davis et al. (2021).

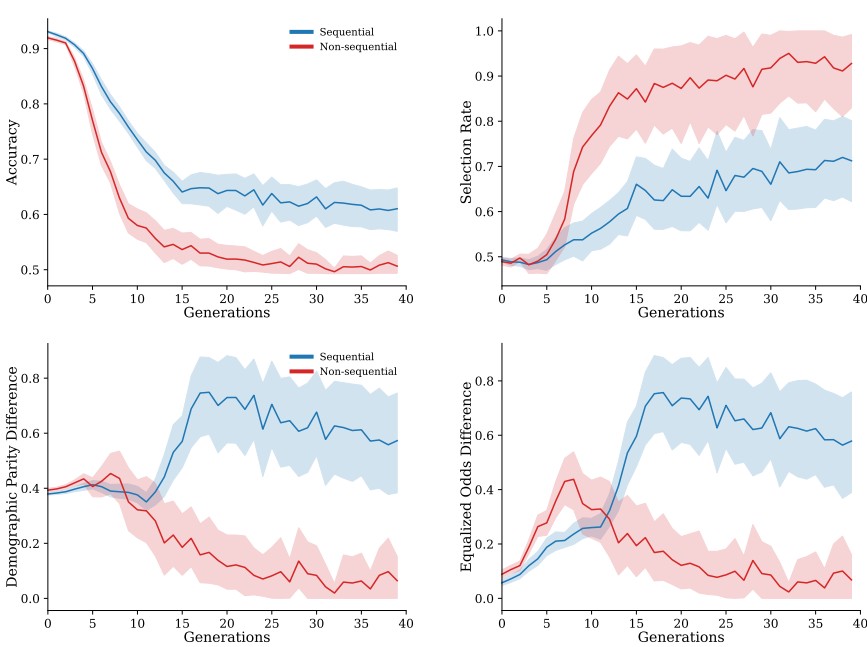

Figure 12: `ColoredMNIST` results for sequential versus non-sequential classifiers in SeqGenSeqClass and SeqGenNonSeqClass.

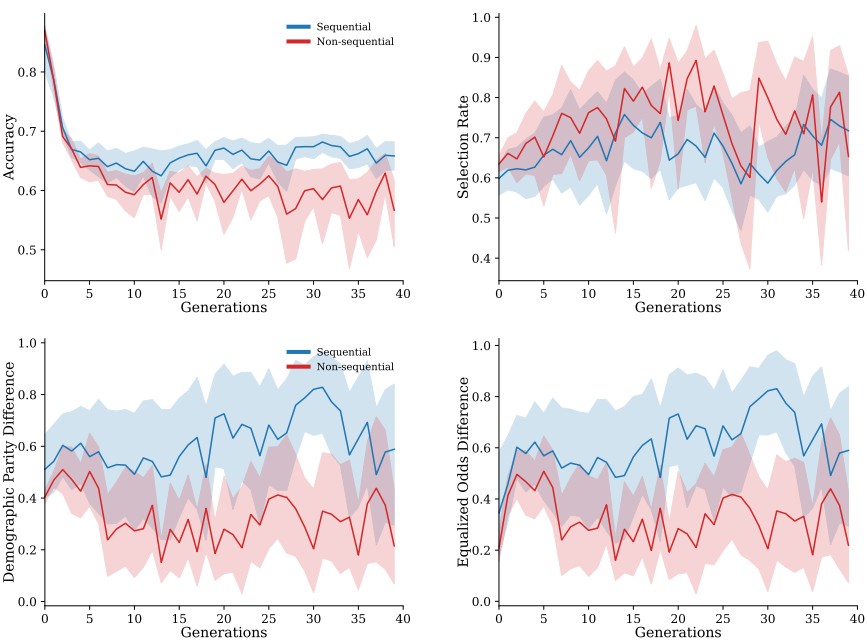

Figure 13: `ColoredSVHN` results for sequential versus non-sequential classifiers in SeqGenSeqClass and SeqGenNonSeqClass.

# F ADDITIONAL FIGURES

## F.1 EXTRA VALIDATION RESULTS

Validation set results continue in Figure 16, Figure 17, and Figure 18. Visualization of model collapse samples for `ColoredMNIST` and `ColoredSVHN` in Figure 19. We also show all results for the accuracy difference between groups in Figure 14 and Figure 15. Again, note that good performance on these metrics does not necessarily indicate fairness so much as adherence to (likely unfair) base rates in these datasets' ecosystems.

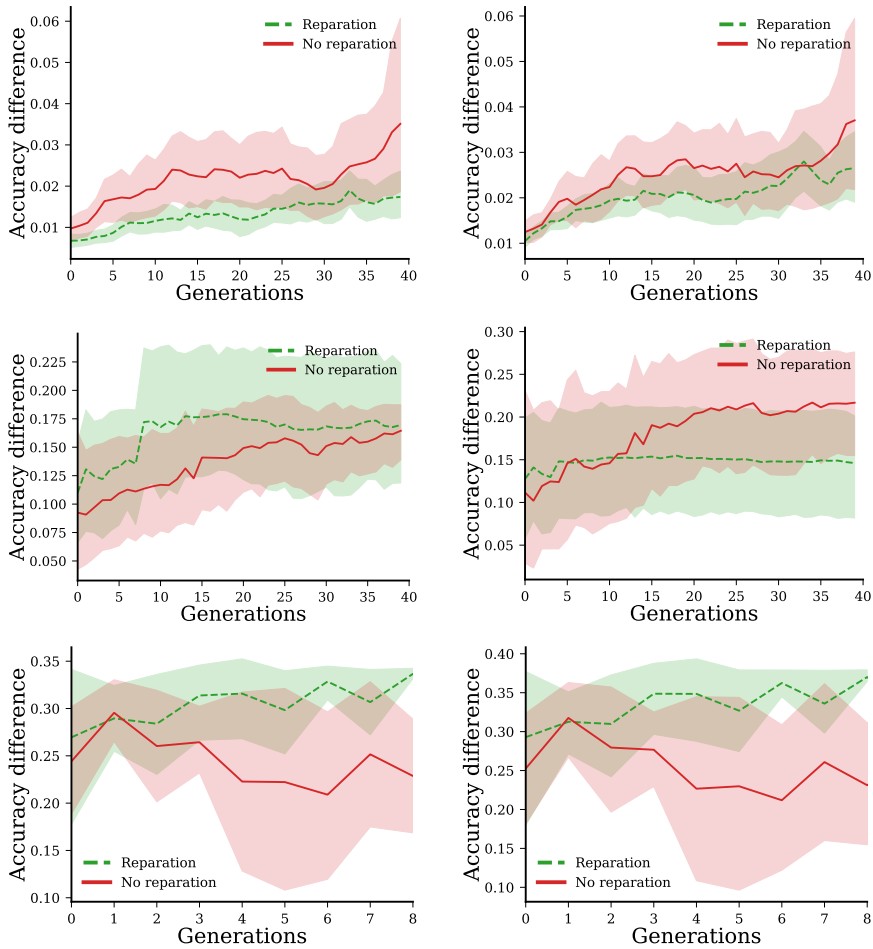

Figure 14: Accuracy difference between groups in SeqClass. Higher difference is worse. AR often incurs significant accuracy cost to achieve better fairness. *Top:* `ColoredMNIST`. *Middle:* `ColoredSVHN`. *Bottom:* `CelebA`. *Left:* Validation set results. *Right:* Test set results.

## F.2 TEST DISTRIBUTION RESULTS

In Appendix C.1 we describe a held-out testing set for `ColoredMNIST` and `ColoredSVHN` which lacks the skew given to bias the training and validation set. The validation set results are presented in the main body. Here, we provide the testing set results for `ColoredMNIST` (Figures 22, 23), `ColoredSVHN` (Figures 24, 25), and for `CelebA` (Figures 20, 21) (for completeness).

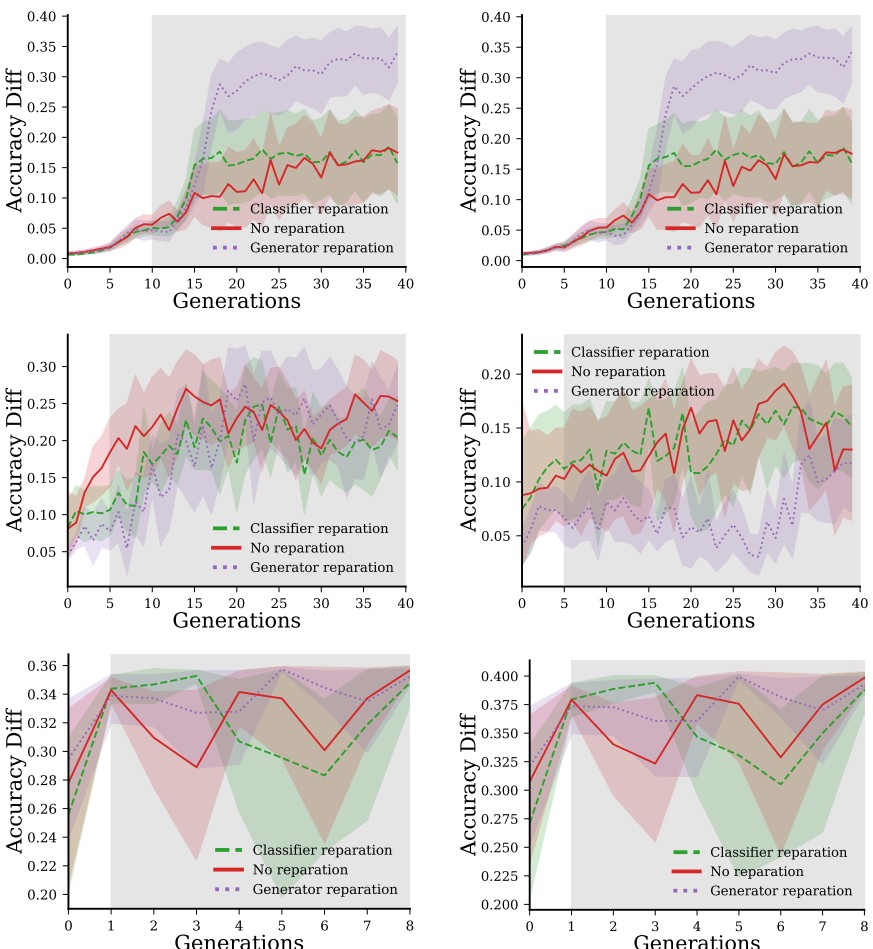

Figure 15: Accuracy difference between groups in `SeqGenSeqClass`. Higher difference is worse. Note that for `ColoredMNIST` generator-side AR incurs significant accuracy cost, as shown in Figure 18 to achieve better group and class representation. *Top:* `ColoredMNIST`. *Middle:* `ColoredSVHN`. *Bottom:* `CelebA`. *Left:* Validation set results. *Right:* Test set results.

# G    RELATIVE PERFORMANCES OF MIDS

If the model trainer is unaware of MIDS occurring over time, they may see only the relative performances (*i.e.* performance of generation $i$ measured *w.r.t.* generation $i - 1$) of each generation compared to its prior generation. In this case, when each generation of models is trained to have relatively high performance, it may look as though the models are performing well, though not when compared to the original data distribution. This may lead to overstating the model's performance, which for the FML metrics results in fairwashing the model due to inadequate validation and testing (Aivodji et al., 2019). For `ColoredMNIST` and `ColoredSVHN`, we report results on the testing set for the 'actual' results (classifiers measured against the testing set) and for the relative results (classifiers measured against the previous generation's classifier predictions on the testing set inputs). We choose not to present these two graphs on the same plots to prevent confusion as they measure two different properties.

For reference, `ColoredMNIST` plots are in Figure 26 for `SeqClass` and Figure 29 for `SeqGenSeqClass`. `ColoredSVHN` plots are in Figure 28 for `SeqClass` and Figure 30 for `SeqGenSeqClass`. We also show results with AR in `SeqClass` for `ColoredMNIST` in Figure 27.

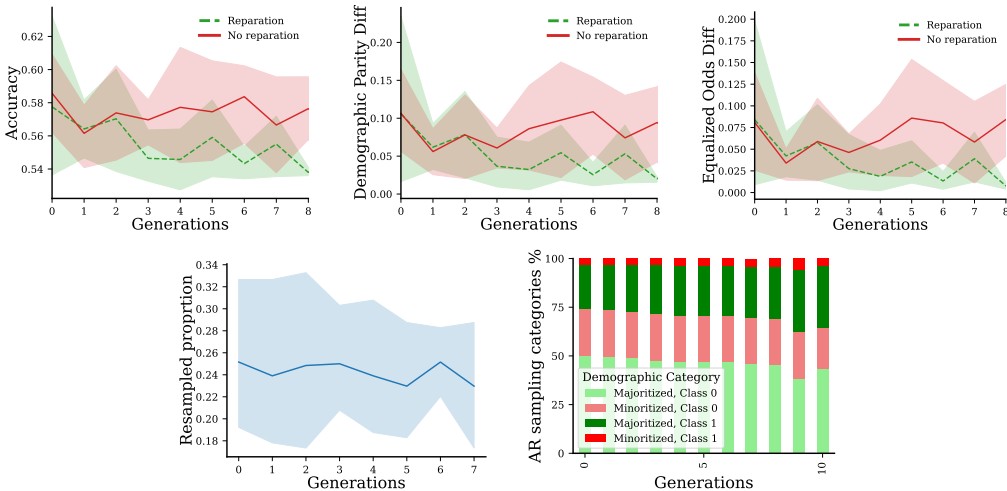

Figure 16: `CelebA` results for **SeqClass** on the validation set. *Top:* accuracy, demographic parity, equalized odds difference. Lower fairness difference is more fair. *Bottom left:* the amount of resampling from $G_0$ due to imbalanced categories after the initial sampling step. *Bottom right:* the category breakdown of sampling $b + r$ times from $G_0$, as described in the initial batch creation step in Section C.4 and Algorithm 1.

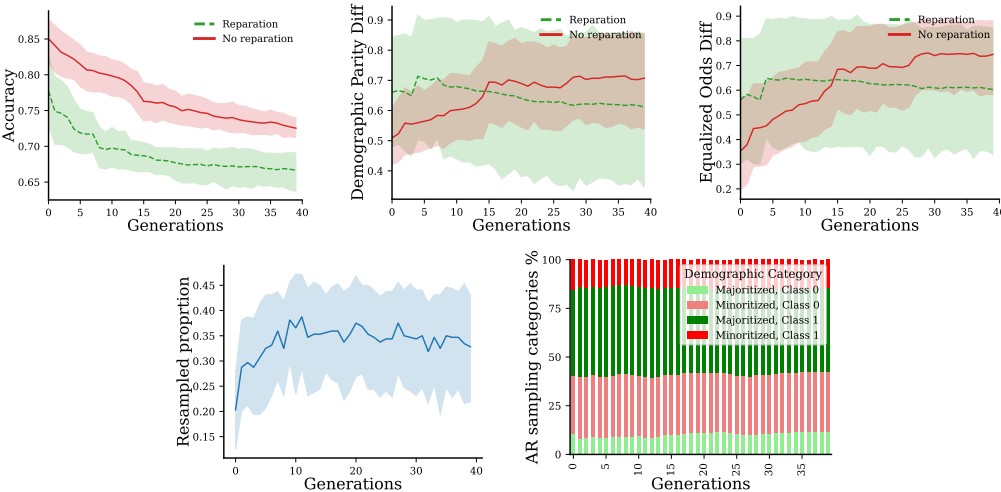

Figure 17: `ColoredSVHN` results for **SeqClass** on the validation set. *Top:* accuracy, demographic parity, equalized odds difference. Lower fairness difference is more fair. *Bottom left:* the amount of resampling from $G_0$ due to imbalanced categories after the initial sampling step. *Bottom right:* the category breakdown of sampling $b + r$ times from $G_0$, as described in the initial batch creation step in Section C.4 and Algorithm 1.

These results also demonstrate how even when training each new classifier with a small tolerance for unfairness can accrue to high unfairness. For example, consider the relative equalized odds results in Figure 26 which on average stay below 0.06 for each generation accrue to over 0.2.

Figure 29 shows the point of model collapse in the **SeqGenSeqClass** setting in `ColoredMNIST` (collapse by generation 15) can be seen in the relative accuracy plot and in the increase in variance in the other relative plots. Similarly as found in the **SeqClass** plots discussed above, low relative equalized odds and a relatively balanced minoritized group under-report the actual unfairness and imbalance.

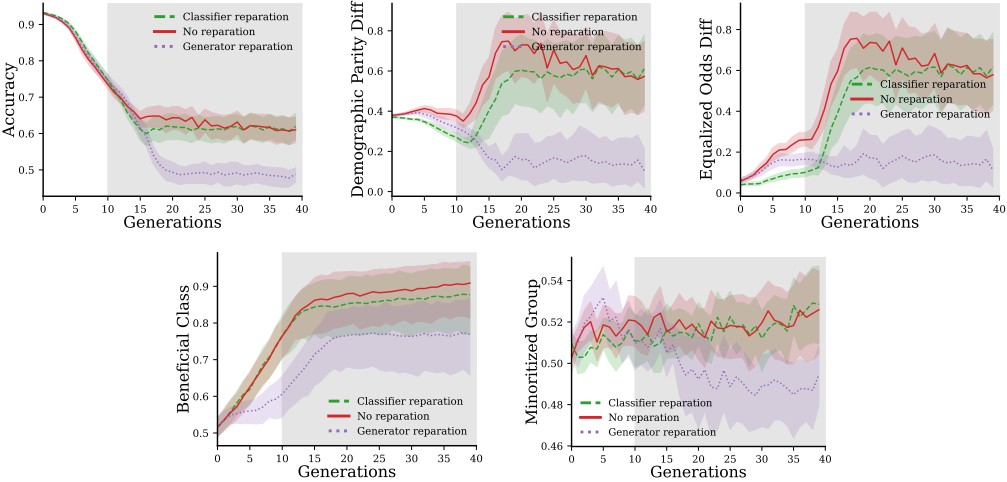

Figure 18: `ColoredMNIST` results for **SeqGenSeqClass** on the validation set. *Top:* shows accuracy and FML metrics (lower is better). Shading shows collapsed generations. *Bottom:* beneficial class and minoritized group representation.

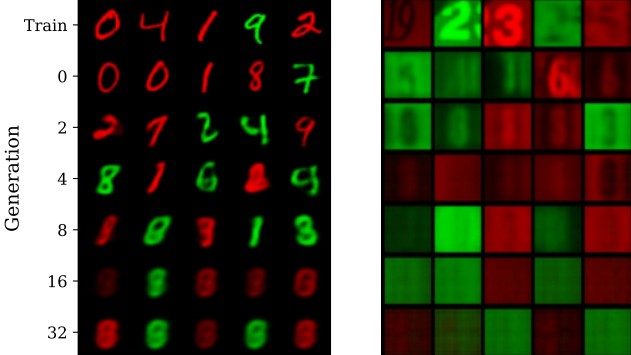

Figure 19: Samples from generators undergoing model collapse in **SeqGenSeqClass** for `ColoredMNIST` (*left*) and `ColoredSVHN` (*right*). Note the first row contains samples from the training set.

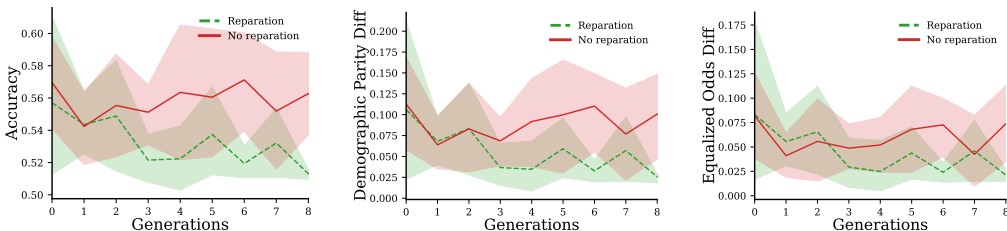

Figure 20: `CelebA` results for **SeqClass** on the testing set. Accuracy, demographic parity, equalized odds difference. Lower fairness difference is more fair.

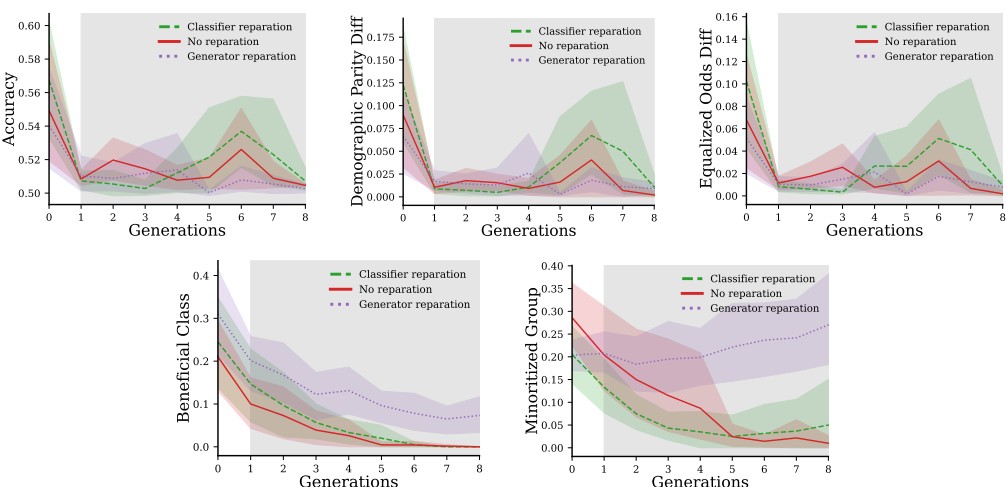

Figure 21: `CelebA` results for **SeqGenSeqClass** on the testing set. *Top:* shows accuracy and FML metrics (lower is better). Shading shows collapsed generations. *Bottom:* beneficial class and minoritized group representation. Note that the beneficial class' is a minority population in the original dataset, and decreases here.

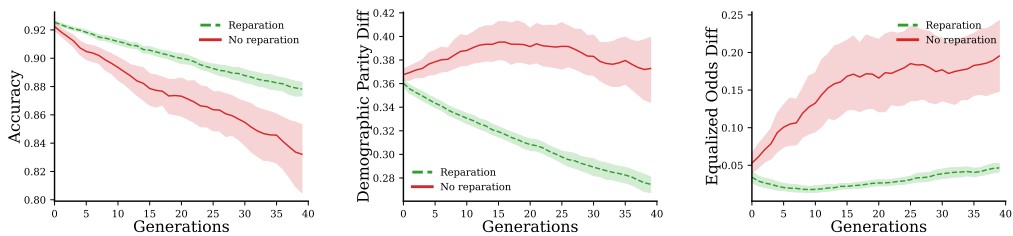

Figure 22: `ColoredMNIST` results for **SeqClass** on the testing set. Accuracy, demographic parity, equalized odds difference. Lower fairness difference is more fair.

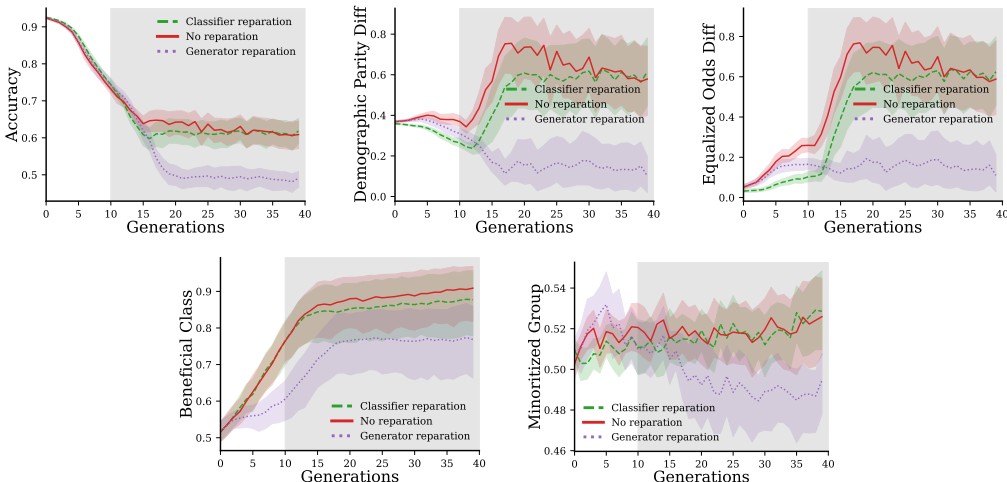

Figure 23: `ColoredMNIST` results for **SeqGenSeqClass** on the testing set. *Top:* shows accuracy and FML metrics (lower is better). Shading shows collapsed generations. *Bottom:* beneficial class and minoritized group representation.

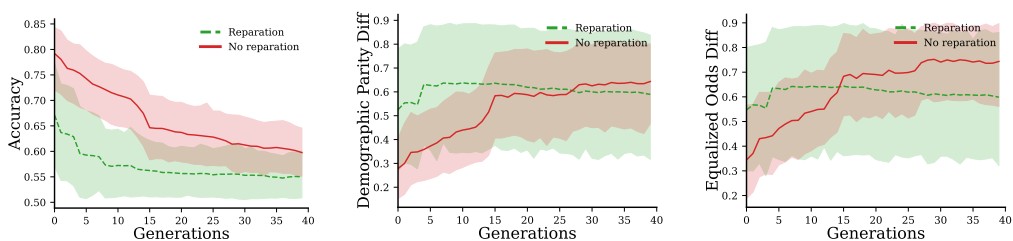

Figure 24: `ColoredSVHN` results for **SeqClass** on the testing set. Accuracy, demographic parity, equalized odds difference. Lower fairness difference is more fair.

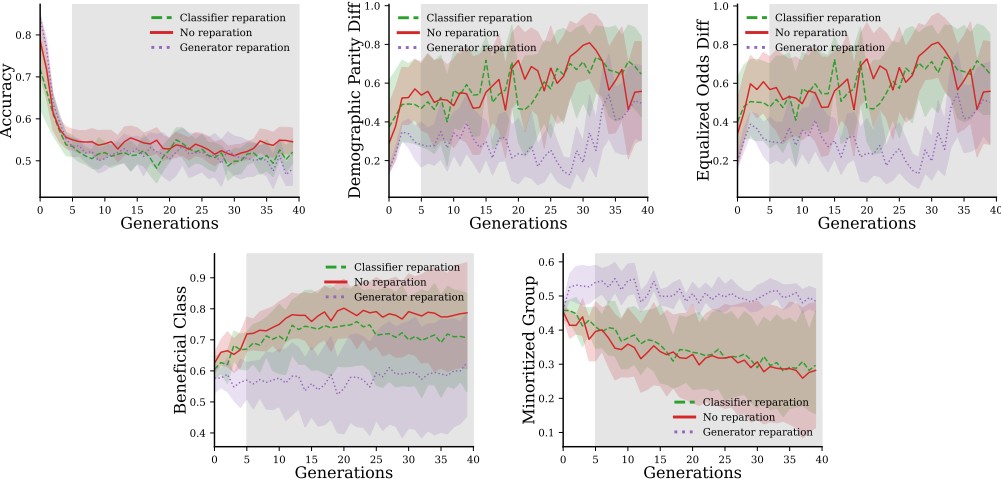

Figure 25: `ColoredSVHN` results for **SeqGenSeqClass** on the testing set. *Top:* shows accuracy and FML metrics (lower is better). Shading shows collapsed generations. *Bottom:* beneficial class and minoritized group representation.

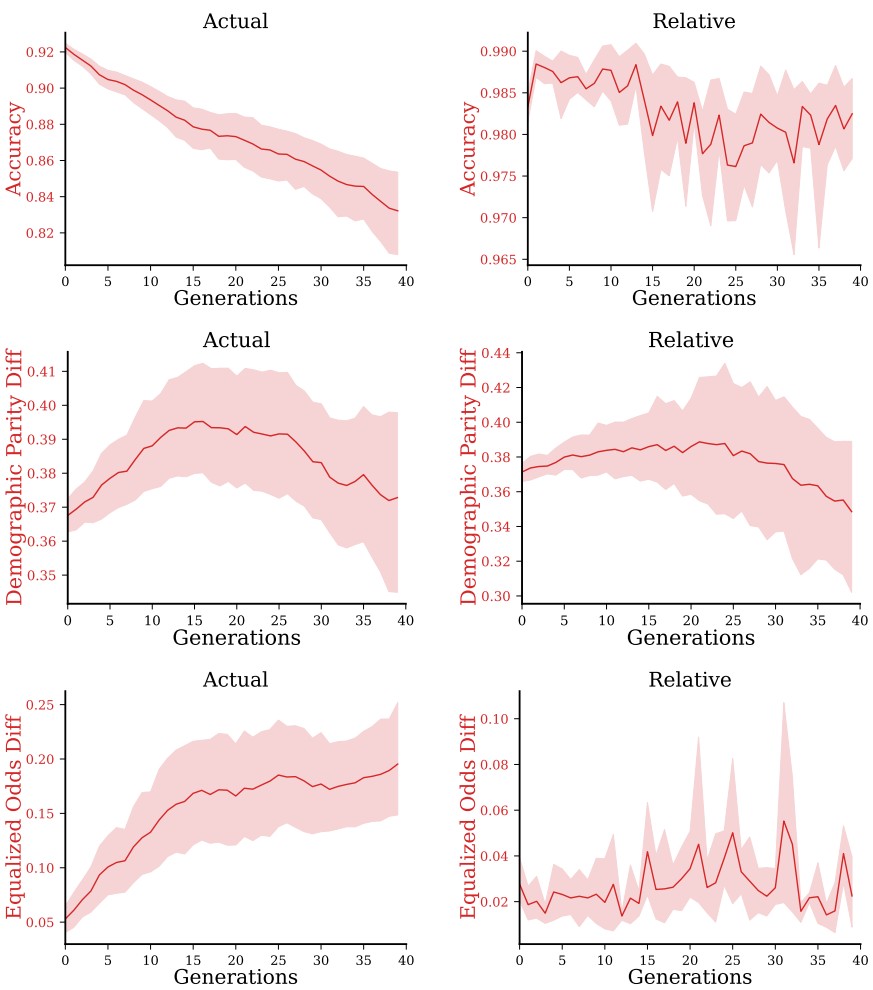

Figure 26: Accuracy, demographic parity difference, and equalized odds difference in SeqClass on ColoredMNIST. Higher accuracy is better, but for the FML metrics higher difference is worse. *Left:* Results presented in the main body. *Right:* Relative performances between models. The model quality of accuracy and equalized odds in the relative performances is far higher than the actual results. In equalized odds, this shows that even if small unfairnesses were tolerated over while training each classifier, the result over time accrues high unfairness compared to the original testing set.

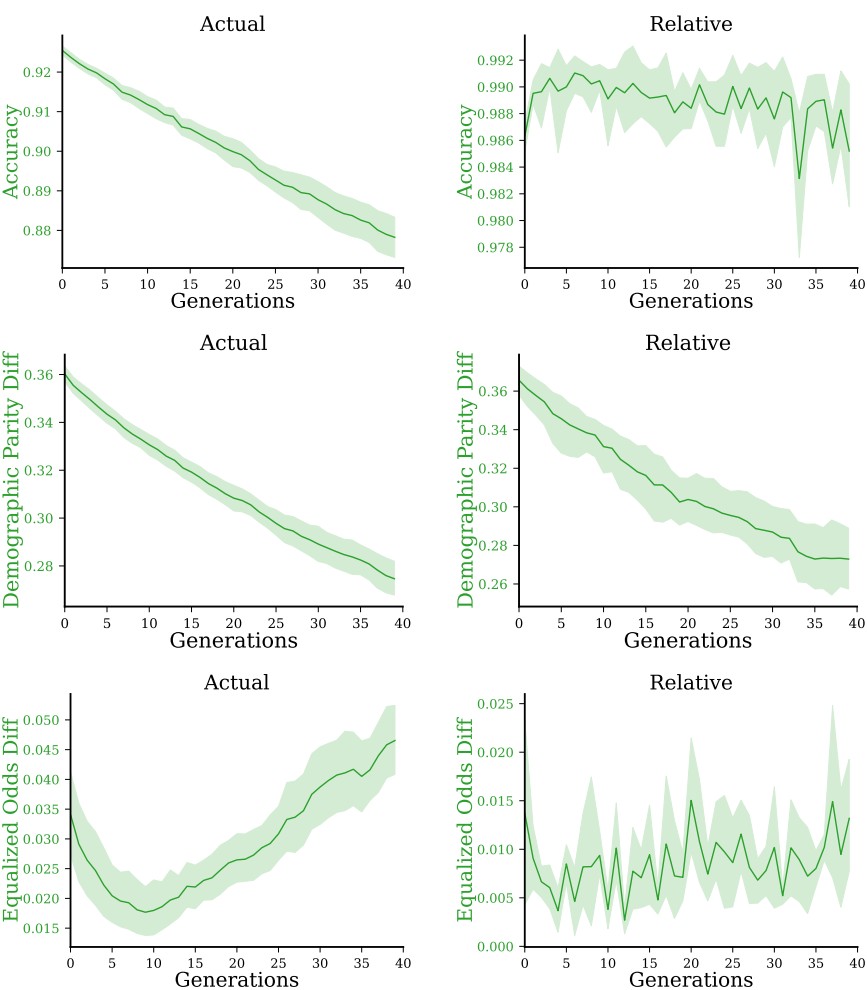

Figure 27: Accuracy, demographic parity difference, and equalized odds difference in SeqClass with classifier-side AR on ColoredMNIST. Higher accuracy is better, but for the FML metrics higher difference is worse. *Left:* Results presented in the main body. *Right:* Relative performances between models.

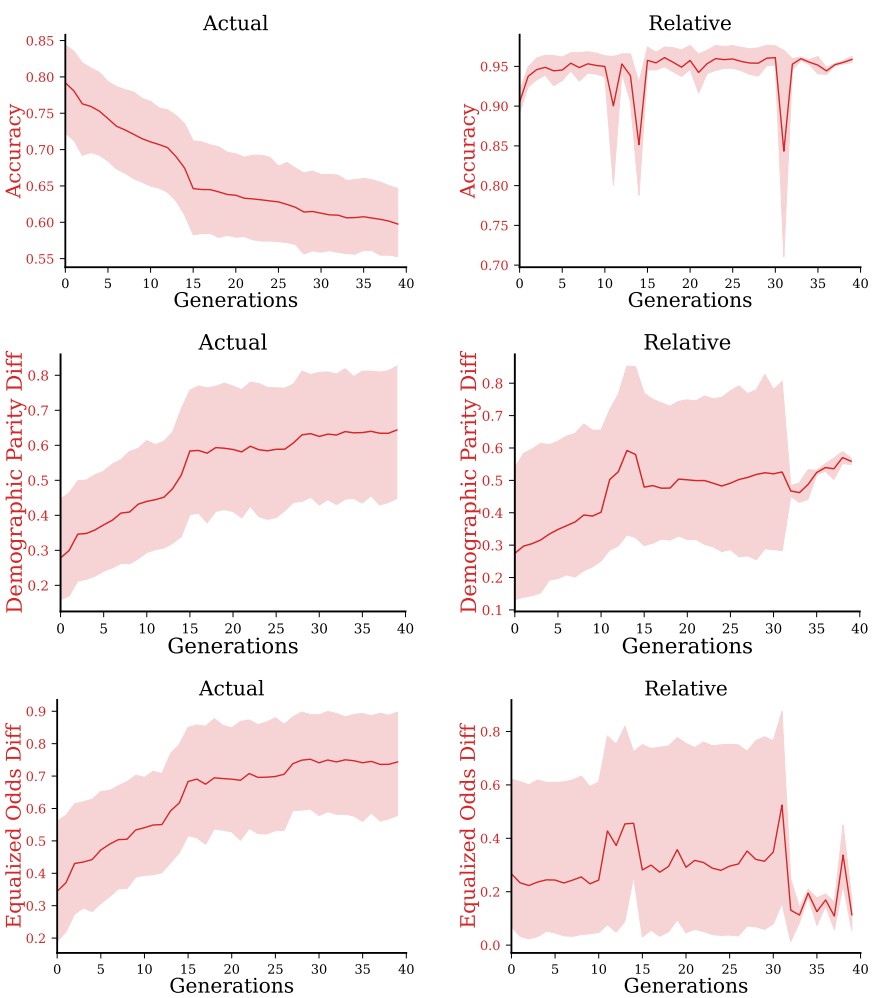

Figure 28: Accuracy, demographic parity difference, and equalized odds difference in SeqClass on ColoredSVHN. Higher accuracy is better, but for the FML metrics higher difference is worse. *Left:* Results presented in the main body. *Right:* Relative performances between models.

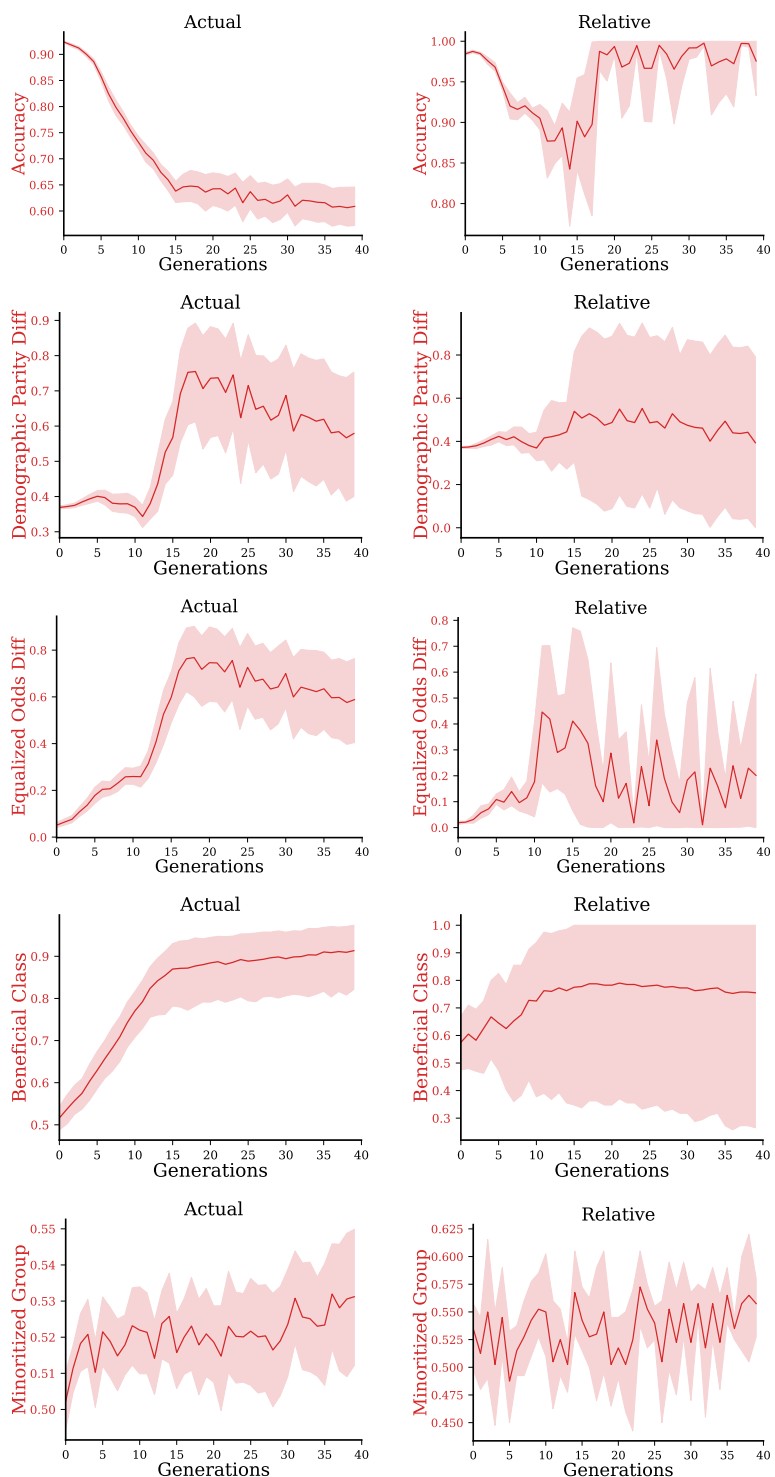

Figure 29: Accuracy, demographic parity difference, equalized odds difference, and rates of the beneficial class and minoritized group in SeqGenSeqClass on ColoredMNIST. Higher accuracy is better, but for the FML metrics higher difference is worse. *Left:* Results presented in the main body. *Right:* Relative performances between models.

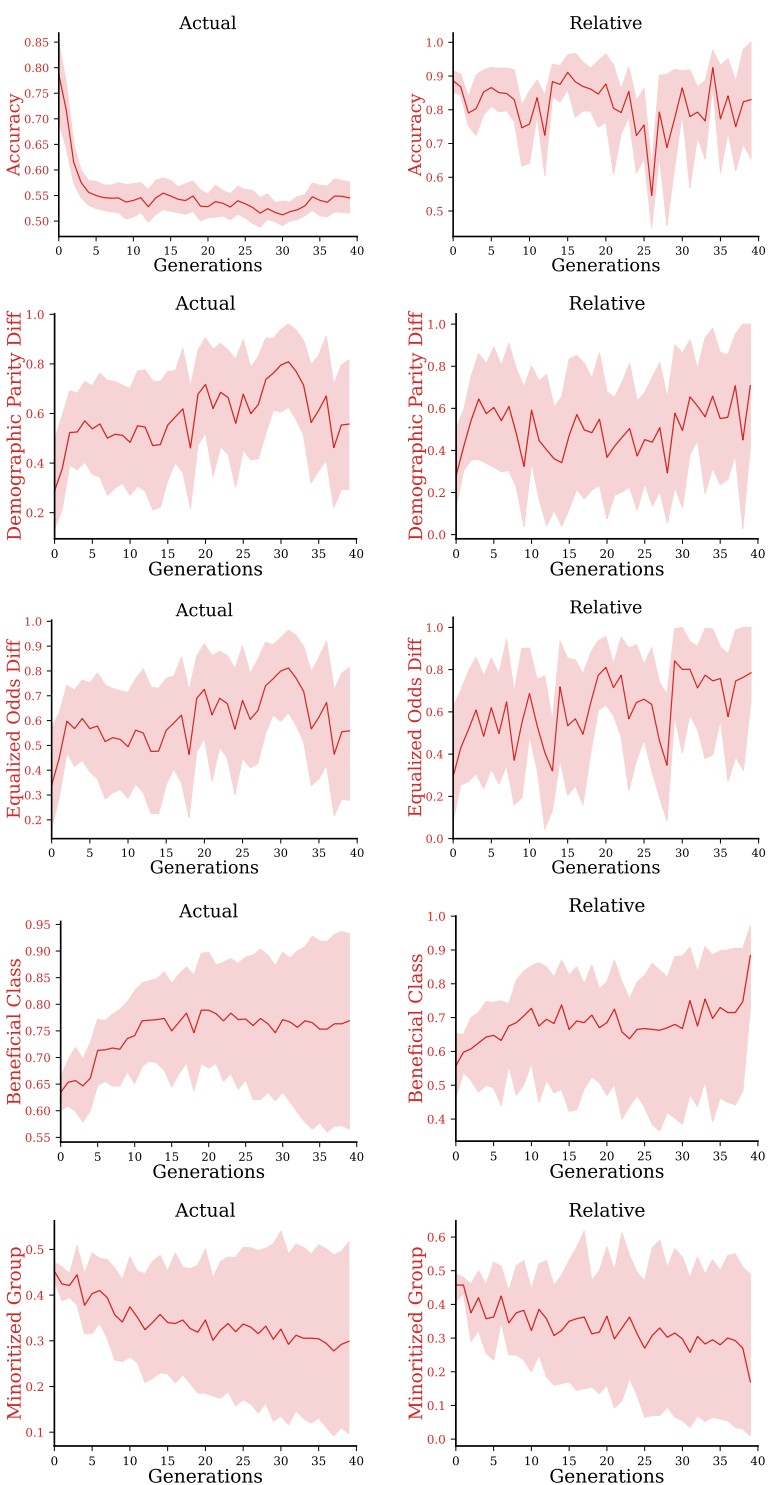

Figure 30: Accuracy, demographic parity difference, equalized odds difference, and rates of the beneficial class and minoritized group in SeqGenSeqClass on ColoredSVHN. Higher accuracy is better, but for the FML metrics higher difference is worse. *Left:* Results presented in the main body. *Right:* Relative performances between models.

