# OpenReview forum: "Fairness Feedback Loops: Training on Synthetic Data Amplifies Bias"
_ICLR.cc/2024/Conference — Submitted to ICLR 2024_

### Official Review · Reviewer_HHFw · 2023-10-24

**Soundness:** 2 fair
**Presentation:** 2 fair
**Contribution:** 2 fair
**Rating:** 3
**Confidence:** 2

**Summary:**

This paper introduces a novel term, "model-induced distribution shift," aiming to encompass various distribution shifts within a single framework. It delves into two scenarios to highlight their effects. Furthermore, the study reveals that model-induced distribution shifts can rapidly result in suboptimal performance, skewed class distribution, and underrepresentation of marginalized demographic groups. To address this challenge, this paper showcases the potential of algorithmic reparation to reduce disparities among sensitive groups.

**Strengths:**

* This paper systematically categorizes the existing literature on model-induced distribution shift, offering a consolidated overview for the machine learning fairness community.
* The experimental setups involving sequences of classifiers and generators are interesting, as they aptly simulate the real-world data distribution shifts induced by deployed models.
* Numerical experiments show the effectiveness of applying algorithmic reparation for mitigating the model-induced data distribution issues.

**Weaknesses:**

I have two main concerns about this work
* In this study, the authors employ a synthetic process using a series of classifiers and generation models to emulate real-world data distribution shifts. How do the authors substantiate that this accurately reflects actual distribution shift behaviors in the real world?
* Could the authors delve into a discussion regarding how current methodologies address the challenges of model-induced distribution shifts?
* Furthermore, the numerical experiments focus solely on a comparison between scenarios with and without algorithmic repair. Could the authors also present comparisons against established baseline methods?

Overall, I'm uncertain about the adequacy of the contribution presented in this work for acceptance. I would recommend that the authors elaborate more on the distinct contributions of their paper in relation to existing methodologies addressing this issue.

**Questions:**

My questions are provided in the Weakness section.

---

> ### Author Response · Authors · 2023-11-18
> **Response to reviewer HHFw**
>
> We thank the reviewer for their comments and address their questions:
>
> ```
> “How do the authors substantiate that this accurately reflects actual distribution shift behaviors in the real world?”
> ```
>
> All of our settings revolve around a sequence of models trained with datasets that get updated over time (whether by adding synthetic data, synthetic labels, or some mixture of synthetic and ‘real’ data). Concerns of use of synthetic data, whether intentionally or otherwise have already appeared in research and media. Our work seeks to better understand MIDS and opportunities for algorithmic reparation. We do not attempt to reflect distribution shift in general, just those induced by models. We also do not study what the impact of MIDS alongside other types of distribution shift might entail, though we do explore interactions between MIDS.
>
> ```
> “Could the authors delve into a discussion regarding how current methodologies address the challenges of model-induced distribution shifts?”
>
> “Could the authors also present comparisons against established baseline methods?”
> ```
> Points 2 and 3 ask for a discussion and experimental comparison against related methods. We show one comparison in Appendix E.2, where we provide an ablation study for the amount of synthetic data present at each generation. We also anticipate that training with fairness at each generation may be insufficient to stop unfairness accumulating due to MIDS (see our response to reviewer QqzD, point 4.b). Otherwise, please refer us to works that can address MIDS.

---

### Official Review · Reviewer_QqzD · 2023-11-01

**Soundness:** 2 fair
**Presentation:** 1 poor
**Contribution:** 2 fair
**Rating:** 3
**Confidence:** 3

**Summary:**

This paper proposed a framework called model-induced distribution shifts (MIDS) that unify several existing notions such as model collapse, unfairness feedback loops, class imbalance, label/ input drift, etc.

**Strengths:**

1.	The formulation and procedure to observe and evaluate MIDS is clear. The flowcharts in Section 3.1 and Section 3.2 are really helpful
2.	The discussions on model collapse for generative models and the performative prediction in Section 4.2 are inspiring

**Weaknesses:**

1.	This paper attempts to encompass several issues such as model collapse, performative prediction, unfairness feedback loops, and algorithmic reparation. However, the benefit and motivation for the unifying MIDS is not clear to me. It is encouraged that the authors clearly state what addition challenge could be solved, or what existing challenges could be better solved by the MIDS framework.
2.	There are existing algorithms that could solve unfairness feedback loops, class imbalance, etc. However, the MIDS framework is not compared with those existing benchmarks.
3.	Algorithmic reparation (AR) should be the most important concept/baseline in this paper; however, it is not technically introduced in the paper. It is encouraged that the authors add more discussion, use case, and operational meaning of AR.
4.	The methodologies proposed in Section 3 for MIDS lack theoretical analysis, performance guarantee, etc.

**Questions:**

1.	In Figure 4, as the number of generations increase, the accuracy drops from 92% to 82%, and the fairness metrics like DP and EO are still large. Why a reduction of accuracy and unfairness will occur at the same time?
2.	Why we need the settings of sequences of generators?

For other questions, please refer to the Weaknesses. I will consider raising the scores if the authors could adequately address my questions in the rebuttal.

---

> ### Author Response · Authors · 2023-11-18
> **Response to reviewer QqzD**
>
> We thank the reviewers for their comments and address their questions below, followed by addressing the weaknesses.
>
> ```
> In Figure 4, as the number of generations increases, the accuracy drops from 92% to 82%, and the fairness metrics like DP and EO are still large. Why does a reduction of accuracy and unfairness occur at the same time?
> ```
>
> Figure 4 depicts sequential classifiers undergoing performative prediction. In the non-reparative results, there are no fairness interventions and we do not expect the classifiers to achieve lower unfairness. However, each generation’s mistakes continue to compound, lowering the accuracy. The algorithmic reparation is successful in eventually yielding batches with even\ideal class and group representation (Figure 4, bottom row), benefiting the fairness metrics. However, as these even batches no longer match the original distribution’s imbalances, there must be some incorrect classification when compared to the original distribution, which is why the accuracy still decreases. Please also see our response to reviewer qqNH’s first question for more information.
>
> ```
> “Why do we need the settings of sequences of generators?”
> ```
>
> The settings with sequential generators are motivated as generated content is published online and re-scraped to form new datasets. For example, in the second paragraph of the introduction we discuss the proliferation of AI-generated baby peacock photos which (as of writing) frequently show up in the first page of Google image search results (as found by Shah and Bender (2022)). A generative model trained on these images would misrepresent the baby peacocks. Another example is from Venugopal et al. (2011), when Google researchers realized that training on previous translations published online might harm future versions of their model. Similarly, Veselovsky et al (2023) showed that AI is now being used by MTurkers, suggesting that we already may have a significant proportion of AI labels in our crowdsourced datasets.
>
> Please see our next comment which addresses the weaknesses.

---

> > ### Author Response · Authors · 2023-11-18
> > **response cont.**
> >
> > Weaknesses:
> >
> > * `However, the benefit and motivation for the unifying MIDS is not clear to me. It is encouraged that the authors clearly state what additional challenges could be solved, or what existing challenges could be better solved by the MIDS framework.` As discussed in our response to reviewer vnkU02, our unification of so-far disparate model-induced effects under MIDS allows us to study them in isolation or when they interact, as in SeqGenSeqClass. It also allows us to work towards fair interventions using algorithmic reparation, and motivate work addressing these problems such as provenance for data, models, and outputs.
> > * `There are existing algorithms that could solve unfairness feedback loops, class imbalance, etc. However, the MIDS framework is not compared with those existing benchmarks.`
> > 	* So far, we have found little related work addressing multiple MIDS. Please refer us to specific related works we may compare against.
> > 	* Since submitting our work, we measured the relative performances of each classifier with respect to its predecessor (instead of the original distribution). For equalized odds, where the ground truth label is assigned by the classifier’s predecessor, we found that the results presented in the main body occur even when there is only a small level of unfairness between each generation. In the relative equalized odds behind the ColoredMNIST results in Figure 4, for example, fairness violations of 0.05 compound to unfairness of 0.2 when compared to the original distribution. This demonstrates that even if we trained each generation’s classifier to be fairly compared to its predecessor, there may still be enough unfairness accumulating over time to cause high unfairness. These results have been added as Appendix G.
> > * `It is encouraged that the authors add more discussion, use case, and operational meaning of AR.`
> > Algorithmic Reparation is not necessarily a technical framework for achieving fairness, but encompasses how to build or adjust algorithms for reparative purposes, using the algorithm to intervene in and correct injustice. Therefore, the technical implementation and objectives of AR are difficult to gauge as they will be heavily context-dependent. Please see So et al. (2022) for an example of how recourse can be used as algorithmic reparation in US housing discrimination.
> > * `The methodologies proposed in Section 3 for MIDS lack theoretical analysis, performance guarantee, etc.`
> > If interested in theoretical backgrounds behind MIDS, please see Shumailov et al. (2023) for a formulation of the SeqGenNoSeqClass setting in terms of a markov chain and Taori and Hashimoto (2023) for a formulation of SeqClass in terms of calibration.
> >
> > [1] Chirag Shah and Emily M. Bender. Situating Search. In Proceedings of the 2022 Conference on Human Information Interaction and Retrieval (CHIIR '22), 2022. Association for Computing Machinery, New York, NY, USA, 221–232.
> >
> > [2] Ashish Venugopal, Jakob Uszkoreit, David Talbot, Franz Och, and Juri Ganitkevitch. Watermarking the Outputs of Structured Prediction with an application in Statistical Machine Translation. In Proceedings of the 2011 Conference on Empirical Methods in Natural Language Processing, 2011.
> >
> > [3] W. So, P. Lothia, R. Pimplikar, A.E. Hosoi, and C. D’Ignazio. Beyond fairness: Reparative algo- rithms to address historical injustices of housing discrimination in the US. In Proceedings of the 2020 Conference on Fairness, Accountability, and Transparency. Association for Computing Machinery, 2022.
> >
> > [4] Ilia Shumailov, Zakhar Shumaylov, Yiren Zhao, Yarin Gal, Nicolas Papernot, and Ross Anderson. The curse of recursion: Training on generated data makes models forget, 2023.
> >
> > [5] Taori, Hashimoto. Data feedback loops: Model-driven amplification of dataset biases. In International Conference on Machine Learning, 2023.
> > [6] Veniamin Veselovsky, Manoel Horta Ribeiro, and Robert West. Artificial Artificial Artificial Intelligence: Crowd Workers Widely Use Large Language Models for Text Production Tasks, 2023.

---

> > > ### Comment · Reviewer_QqzD · 2023-11-22
> > >
> > > I appreciate the authors for their response. I will stand my current score.

---

### Official Review · Reviewer_vnkU · 2023-11-02

**Soundness:** 1 poor
**Presentation:** 1 poor
**Contribution:** 2 fair
**Rating:** 3
**Confidence:** 3

**Summary:**

The paper is about the feedback loop when models are continuously trained on data generated by them. The authors introduce model-induced distribution shifts (MIDS) which occur as previous model outputs pollute new model training sets over generations of models. They provide a taxonomy for MIDS and demonstrate that their fairness effects lead to a lack of representation and per-
formance on minoritized groups within a few model iterations. The authors propose Algorithmic Reparation (AR) as another explicit MIDS
deployed with the goal of reducing societal inequity and correcting for historical oppression; they use AR to reduce the unfairness impacts of other MIDS by sampling for minoritized group representation, leading to better downstream fairness over time.

**Strengths:**

* The feedback loop problem is important and interesting.
* The authors propose a setup in which this problem can be studied and explore Algorithmic Reparation as a possible solution.

**Weaknesses:**

Personally, I find the paper a bit hard to understand.

* Introduction seems a bit verbose and overly lyrical in moments, making it harder to read and follow\
"recent demographic information of the Black population" - it is written that the maps are from 1939 and 1955. I am not sure that this is very recent.

* I am not sure that MIDS require their own "taxonomy", given that there are only label and input drifts (Table 1). Impact of feedback loop in fairness has been acknolwedged as a problem for a while (Mehrabi et al. 2019)

* The related work is not clearly discussed in the paper, making it a bit hard to get the overall context and contributions with respect to prior work.

* A recent related work by Taori and Hashimoto (2023) is missing. How does this paper relate to them?

Mehrabi et al., A Survey on Bias and Fairness in Machine Learning, arXiv 2019\
Taori and Hashimoto , Data Feedback Loops: Model-driven Amplification of Dataset Biases, 2023

**Questions:**

Q1: How do you select the datasets and what are the motivations for them and the experimental setup exactly?

Q2: What are the models that you employ for the classifiers and the generator?

Q3: Why does the accuracy drops over time in Fig. 5 and 6?

---

> ### Author Response · Authors · 2023-11-18
> **Response to reviewer vnkU**
>
> We thank the reviewer for their comments, and address their questions here:
>
> ```
> “Q1: How do you select the datasets and what are the motivations for them and the experimental setup exactly?”
> ```
>
> Dataset choice: We follow a common experimental setting used in computer vision fairness research, adapting datasets into binary classification and binary sensitive grouping (Arjovsky et al. 2022). We use MNIST and SVHN as they both contain the same number of classes. Their difference is in difficulty (SVHN is harder, usually) and class imbalance when split at 5. These two datasets, despite their similarity, showed very different points of model collapse and opposite behavior when observing the accuracy difference between groups (Figures 14 and 15) as ColoredMNSIT reparation resulted in less accuracy difference unlike the other datasets.
> We chose CelebA for a much more complex/real-world dataset with many sensitive features and well-documented disparities present in the dataset’s base rates (see Table 2).
> All of these datasets can either be manipulated (as in Arjovsky et al. 2022) into binary group and binary class classification problems (MNIST, SVHN) or already have these features (CelebA). This allows for ease of evaluation of our metrics and allows us to ablate both group and class imbalance to study the mechanics of MIDS.
> Experimental setup details may be found in Appendix C, we also added further discussion for the reasons we chose these datasets in Appendix C.1.
>
> ```
> “Q2: What are the models that you employ for the classifiers and the generator?”
> ```
>
> Models for classifiers are CNNs or ResNets, and generators are beta-VAEs. Full details are in Appendix C.3, or in our code hosted anonymously https://anonymous.4open.science/r/ FairFeedbackLoops-1053/README.md
>
> ```
> “Q3: Why does the accuracy drop over time in Fig. 5 and 6?”
> ```
>
> The accuracy decreases in Figures 5 and 6 (SeqGenSeqClass) due to model collapse. The accuracy is measured at each generation by inputting a held-out set into the classifier and recording performance. However, each classifier was trained from the distribution represented by a generative model undergoing model collapse. As the generators lose the ability to accurately represent the initial distribution (see Figure 19 to see how MNIST and SVHN digits deform over time), we expect the classifiers to lose their ability to discriminate between classes, and their accuracy should decrease.
>
> Please see our next comment where we address the weaknesses.

---

> > ### Author Response · Authors · 2023-11-18
> > **response cont.**
> >
> > We also comment on the weaknesses:
> >
> > * Thank you for the catch on the `recent demographic information of the Black population,` this is now fixed.
> > * `I am not sure that MIDS require their own "taxonomy", given that there are only label and input drifts (Table 1). Impact of feedback loop in fairness has been acknowledged as a problem for a while (Mehrabi et al. 2019)`
> > While fairness feedback loops are well-known in literature, they have not yet been connected to other phenomena such as model collapse or disparity amplification. We seek to unify these effects as their causes are distinct from traditional forms of distribution shift or domain adaptation. Unifying and recognizing these MIDS together opens avenues to study their co-occurrence/ co-operation, as well as study fairness interventions such as algorithmic reparation. The problems caused by MIDS motivates work in data and model attribution, but also in provenance for both generated data and data annotation (the latter being often overlooked).
> > * `The related work is not clearly discussed in the paper, making it a bit hard to get the overall context and contributions with respect to prior work.`
> > Additional literature on fairness in ML and its critiques is discussed in Appendix A, and descriptions of known MIDS are provided in Appendix B. Related work so far focuses on describing a MIDS, exploring its effects, and perhaps establishing some theoretical bounds on it. An additional related work previously missing from our paper is discussed below.
> > * `A recent related work by Taori and Hashimoto (2023) is missing. How does this paper relate to them?`
> > * * We were unaware of this work until shortly before reviews were released, and have since added a discussion and citation in Appendix B.3. Taori and Hashimoto (2023) find a theoretical upper-bound for amplification of biases due to model feedback, applicable to our SeqGenNoSeqClass and SeqClass settings. They measure bias amplification as the difference between the original dataset/distribution bias and the bias of the current generation, and propose two interventions: overfitting models and/or allowing for a mixture of synthetic and human-generated data. Both of these methods reduce the bias amplification by encouraging each model to better/more closely represent the original distribution over time.
> > * * Our work departs from theirs in three main regards. Firstly, we primarily study their worst-case scenario of entirely model-generated training sets (though we ablate the amount of synthetic data in Appendix E.2). Secondly, we advocate for using models to intervene in unfair systems through Algorithmic Reparation. In our experiments, we implement AR via a sampling scheme that attempts to achieve a “fair ideal” which may be very different from the rates of the original distribution (consider the base-rate disparities of CelebA, for example). Thirdly, we expand their investigation by observing the effects of co-occurring MIDS in the SeqGenSeqClass setting, finding an interesting dynamic where the classifiers undergoing performative prediction keep up with the changes in their input distribution arising from model collapse.
> >
> > [1] Martin Arjovsky, Leon Bottou, Ishaan Gulrajani, and David Lopez-Paz. Invariant risk minimization, 2020. URL https://arxiv.org/abs/1907.02893
> >
> > [2] Taori, Hashimoto. Data feedback loops: Model-driven amplification of dataset biases. InInternational Conference on Machine Learning 2023.

---

> > > ### Comment · Reviewer_vnkU · 2023-11-22
> > >
> > > I thank the authors for their efforts and rebuttal.
> > >
> > > While the work and the presented ideas have their merit, I still believe that the technical contributions, the theoretical framework and the experimental setup are not yet clear enough (to me). Moreover, beta-VAEs for example are very crude generative models, which makes me wonder how realistic, practical and informative the evaluation is.
> > >
> > > After reading the rest of the reviews, I would like to keep my score.

---

### Official Review · Reviewer_qqNH · 2023-11-04

**Soundness:** 2 fair
**Presentation:** 3 good
**Contribution:** 2 fair
**Rating:** 3
**Confidence:** 3

**Summary:**

This paper provides a conceptual taxonomy of various ways in which repeated training of a model (either a classifier or a generator) in the same data ecosystem lead to negative effects on fairness such as degradation in fairness metrics and distorted class proportions. The overall phenomenon is terms MIDS - Model Induced Distribution Shift, and encompasses previously studied phenomena/concepts such as model collapse, performative prediction and fairness feedback loops. Using experiments on a sequence of recursively trained classifiers (and likewise generators), the paper exposes the several possible harmful effects of MIDS. The paper then introduces a simple resampling scheme for Algorithmic Reparation (AR) into the MIDS framework, and shows through experiments that the proposed AR can ameliorate some of the harmful effects of MIDS.

**Strengths:**

- The paper addresses the important issue of studying the fairness effects of repeated training in a data ecosystem.
- The paper presents a clear model of sequentially training classifiers and generators that can be used to simultaneously model various effects like feedback loops, performative effects and model collapse.

**Weaknesses:**

- The paper does not propose a mathematical model for the proposed MIDS scheme. Although various effects of retraining are shown through experiments, there is no theoretical investigation on the root causes of these effects, or the efficacy of the presented Algorithmic Reparation (AR) scheme.
- The experimental results with the proposed AR scheme based on resampling are not very convincing. For example, the decrease in the equalized odds difference with reparation in Figure 4 is not monotonic. The gap closes for a few generations and then seems to rise up again. The plots on fairness metrics in Figure 5 are even less convincing.

**Questions:**

- What is the reason for the decrease in the equalized odds difference with reparation in Figure 4 not being monotonic?
- The paper claims that "Generator-side AR improves fairness and minoritized representation", but plots on fairness metrics in Figure 5 do not corroborate this claim. Please explain.
- I do not understand the claim, "Performative prediction on model collapse leads to higher utility". Please explain what you mean by this.

---

> ### Author Response · Authors · 2023-11-18
> **Response to reviewer qqNH**
>
> We thank the reviewer for their comments, and address their questions here.
>
> ```
> What is the reason for the decrease in the equalized odds difference with reparation in Figure 4 not being monotonic?
> ```
> Equalized odds is the maximum of the difference between the true positive rate (TPR) difference or false positive rate (FPR) difference between groups, and is therefore dependent on the ground truth. In Figure 4, we show our metrics as measured with respect to the original distribution (see Table 2), where there is correlation between the sensitive attribute value and class. The first classifiers in early generations start off unfair wrt the original distribution. As AR starts to balance the training set, the classifiers achieve a minimal unfairness wrt the original distribution (generation 13). However, as AR continues to balance the dataset and reach its ideal (Figure 4 bottom right), in the view of the original distribution the minoritized group is being “unfairly” benefited over the majoritized group, which is why the unfairness increases. When plotted, AR causes the majoritized TPR to decrease until it is lower (by about 0.035 also the final EOdds at generation 40) than the minoritized TPR, showing the “unfair” benefit AR gives to the minoritized group when seen from the view of fairness in the original distribution. There is also the presence of label shift due to the sequential classifiers, but these effects taken together is why we mention that we might not want to achieve fairness in Figures 4-6.
>
> ```
> The paper claims that "Generator-side AR improves fairness and minoritized representation", but plots on fairness metrics in Figure 5 do not corroborate this claim.
> ```
> We find that using the generator-side AR technique does improve fairness and minoritized representation. In Figure 5, this is shown by the purple dotted line where the demographic parity and equalized odds values are lower (fairer) than (a) classifier-side reparation and (b) results without reparation. The generator-side reparation maintains the original minoritized group population at 50% of the total, as in the ColoredMNIST dataset. Because classifier-side AR and baseline results diminish this population, they result in less representation.
>
> ```
> I do not understand the claim, "Performative prediction on model collapse leads to higher utility". Please explain what you mean by this.
> ```
> This claim means that the combined setting SeqGenSeqClass allows model collapse (MIDS on the generative models) and performative prediction (MIDS on the classifiers) to co-exist. Compared to the SeqGenNoSeqClass (i.e., just model collapse), this setting has higher utility (by this we mean accuracy). This result highlights cooperation between the model collapse and performative prediction MIDS as the SeqGenSeqClass classifiers adapt to changes due to model collapse. Please see Appendix E.3 for further discussion.
>
> In retrospect, the wording of this claim is confusing so we amended it to “Performative prediction combined with model collapse yields higher accuracy.”
>
> We also address the weaknesses:
> * `The paper does not propose a mathematical model for the proposed MIDS scheme.` If interested in theoretical backgrounds behind MIDS, please see Shumailov et al. (2023) for a formulation of the SeqGenNoSeqClass setting in terms of a Markov chain and Taori and Hashimoto (2023) for a formulation of SeqClass in terms of calibration.
> * `the decrease in the equalized odds difference with reparation in Figure 4 is not monotonic. The gap closes for a few generations and then seems to rise up again. The plots on fairness metrics in Figure 5 are even less convincing.` Question about Figure 4 is answered above in point 1. The AR scheme as presented in Figure 5 shows that generator-side reparation, while performing with less accuracy, maintains class and group balance and achieves far lower unfairness than the results without any reparation.
>
> [1] Ilia Shumailov, Zakhar Shumaylov, Yiren Zhao, Yarin Gal, Nicolas Papernot, and Ross Anderson. The curse of recursion: Training on generated data makes models forget, 2023.
>
> [2] Taori, Hashimoto. Data feedback loops: Model-driven amplification of dataset biases. InInternational Conference on Machine Learning 2023.

---

### Author Response · Authors · 2023-11-18
**General Response**

We thank the reviewers for their comments, and have implemented these in our revision upload. Changes are color-coded: blue for new additions, red for fixes, and orange for shortenings.

---

### Meta-Review · Area_Chair_KUp7 · 2023-12-10

**Metareview:**

The paper attempts to unify several distribution shift effects under one concept called "model-induced distribution shifts" (MIDS). MIDS describes the phenomena where model outputs are fed into a new training set, leading to biases, model collapse, and other unintended consequences. The paper is eloquently motivated and addresses an important aspect of the societal effects of learning models. The connection with algorithmic reparation is particularly compelling.

Despite its important and timely premise, the reviewers agreed on the paper's limitations. These limitations mostly centered around the lack of rigor in definitions provided in the paper and unconvincing numerical results. Having said that, the reviewers also encouraged the authors to continue developing this line of work.

**Justification For Why Not Higher Score:**

As noted across reviews, the paper needs to have a clearer methodological and experimental development prior to acceptance.

**Justification For Why Not Lower Score:**

N/A

---

### Decision · Program_Chairs · 2024-01-16

Reject